# Wastewater surveillance reveals patterns of antibiotic resistance across the United States

Sooyeol Kim [1], Alessandro Zulli[2], Elana M. G. Chan [2], Dorothea Duong[3], Rebecca Y. Linfield [4], Caroline E. M. McCormack [1], Bradley J. White[3,5], Marlene K. Wolfe[6], Alexandria B. Boehm [2] & Amy J. Pickering [1,7,8] ✉

Antibiotic resistance is a growing public health threat, with over 2.8 million antibiotic-resistant infections and 35,000 attributable deaths annually in the U.S. Here, we sought to use wastewater monitoring to assess community-level burden of antibiotic resistance. This study quantifies concentrations of antibiotic resistance genes (ARGs) by digital droplet PCR in wastewater solids obtained from 163 wastewater treatment plants in 40 states in the United States. We measure 11 ARGs that confer resistance to beta-lactams ($bla_{CMY}$, $bla_{CTX-M}$, $bla_{KPC}$, $bla_{NDM}$, $bla_{OXA-48}$, $bla_{TEM}$, $bla_{VIM}$), colistin ($mcr$-$1$), methicillin ($mecA$), tetracycline ($tetW$), and vancomycin ($vanA$). The South has higher overall ARG concentrations compared to the Midwest. We pair these data with national data sets including antibiotic use, social vulnerability, size of animal agriculture operations, location of healthcare facilities, and presence of airports to investigate potential drivers of resistance. We also generate predictive maps of ARG concentrations for counties with data within the range of our training set in the United States. We show social vulnerability indicators (housing burden and access to health insurance) and indicators of international travel are associated with increased ARG concentrations in wastewater, while antibiotic usage is only weakly positively associated. Our results provide a national baseline of ARG concentrations and highlight the complexity of factors driving spread of antibiotic resistance.

Antimicrobial resistance (AMR) is a critical and growing global public health threat, with resistant bacterial infections, such as carbapenem resistant *Enterobacterales* and vancomycin resistant *Enterococcus*, becoming more prevalent[1]. The 2019 Antibiotic Resistance Threats Report by the U.S. Centers for Disease Control and Prevention (CDC) highlights the severity of the crisis, reporting over 2.8 million antibiotic resistant infections annually, resulting in more than 35,000 deaths[2]. This is an underestimate, as it is based on people who seek medical attention. Traditional clinical surveillance for antibiotic resistant infections involves testing bacteria isolated from clinical specimens to assess antibiotic susceptibility. This method is resource-intensive and only reflects antibiotic resistance genes (ARGs) in individuals seeking medical treatment, limiting our understanding of the true burden and diversity of AMR in the broader community.

Various anthropogenic factors contribute to the levels of AMR circulating in communities, but the relative influence of these factors remains unclear. Improper use of antibiotics in both healthcare and agriculture has long been recognized as a major driver of antibiotic

[1]Department of Civil and Environmental Engineering, University of California, Berkeley, CA, USA. [2]Department of Civil and Environmental Engineering, Stanford University, Stanford, CA, USA. [3]Verily Life Sciences, South San Francisco, CA, USA. [4]Division of Infectious Diseases and Geographic Medicine, Department of Medicine, Stanford University, Stanford, CA, USA. [5]Google LLC, Mountain View, CA, USA. [6]Gangarosa Department of Environmental Health, Rollins School of Public Health, Emory University, Atlanta, GA, USA. [7]Biohub, San Francisco, San Francisco, CA, USA. [8]Blum Center for Developing Economies, University of California, Berkeley, Berkeley, CA, USA. ✉e-mail: pickering@berkeley.edu

resistance, with overprescription and misuse accelerating the development of resistant bacteria[3]. However, recent studies have increasingly emphasized the role of socioeconomic factors in shaping resistance patterns[4,5]. Disparities in access to healthcare, sanitation, and education can influence not only antibiotic use but also the effectiveness of infection control measures. In a global analysis, Collignon et al. reported limited association between clinical isolates, antimicrobial resistance levels, and antibiotic consumption but found stronger association with socioeconomic factors such as GDP per capita, education, infrastructure, and public healthcare spending[5]. Similarly, a global analysis of sewage metagenomes by Gupta et al. and Lee et al. identified significant correlations between resistance levels and socioeconomic factors[6,7]. Additionally, global travel has also been found to play a key role in spreading resistant pathogens, facilitating the dissemination of resistance across borders[8,9]. These interconnected drivers highlight the need for a comprehensive examination of multiple determinants of antibiotic resistance.

Recent advances in wastewater monitoring for SARS-CoV-2 and other viral pathogens have established data collection infrastructure that can be leveraged to monitor various markers of infectious disease and human health, including ARGs, in a timely and comprehensive manner[10]. Wastewater has consistently been recognized as a reservoir for ARGs. While previous research has characterized ARG levels and their fate within wastewater treatment systems using PCR-based methods, studies undertaking national-scale community antibiotic resistance surveillance via wastewater monitoring are still emerging. Prior work assessing community-level ARGs in wastewater has largely involved either global spread or differences in the wastewater resistome globally[4,6,11,12], or is limited to single cities or states[13–15]. Additionally, many previous wastewater ARG studies conducted for the purpose of monitoring community antibiotic resistance have used shotgun metagenomics. While metagenomic approaches are useful in understanding the overall ARG diversity, they often lack the sensitivity and quantitative precision needed for accurate abundance estimates. In contrast, digital PCR (dPCR) has the advantage of being highly sensitive and quantitative, allowing for normalized abundance estimates and comparison across samples in time and space[16–18].

In this study, we use droplet digital PCR (ddPCR) to quantitatively measure the abundance of 11 clinically relevant ARGs that confer resistance to beta-lactams ($bla_{CMY}$, $bla_{CTX-M}$, $bla_{KPC}$, $bla_{NDM}$, $bla_{OXA-48}$, $bla_{TEM}$, $bla_{VIM}$), colistin ($mcr-1$), methicillin ($mecA$), tetracycline ($tetW$), and vancomycin ($vanA$) from 163 wastewater sites to generate a cross-sectional data set of the U.S. By linking mostly publicly available secondary data, we investigate potential determinants of wastewater ARG concentrations, including antibiotic use, social vulnerability, the size of animal agriculture operations, and the locations of healthcare facilities and airports. Finally, using a subset of our data as a training set, we construct a random forest model to predict antibiotic resistance gene prevalence in counties across the U.S. that can be interpolated with our training set.

## Results

### Clinically relevant ARGs are at high concentrations in wastewater across the U.S

We measured the concentration of 11 antimicrobial resistance genes (ARGs) across 163 sites in the U.S, spanning 40 states for a cross-sectional data set (Fig. 1A). An average of 2.7 samples were collected per wastewater treatment plant (WWTP) over a week in May of 2024 and concentrations were averaged to obtain one representative concentration for each target per treatment plant (Table S1). We selected the targets based on the Centers for Disease Control Antibiotic Resistance Threats Report that identifies pathogen-resistance combinations as urgent, serious, and concerning based on their threat to human health[1]. Targets included genes that confer resistance to beta-lactams, colistin, methicillin, tetracycline, and vancomycin. The ARGs

$bla_{CMY}$, $bla_{CTX-M}$, $bla_{KPC}$, $bla_{OXA-48}$, $bla_{TEM}$, and $tetW$ consistently exceeded $10^{-7}$ copies per copy of 16S rRNA gene and were detected in all samples (Fig. 1B). Among these, $tetW$ exhibited the highest concentration across all samples, with minimal variation in its distribution. In contrast, $bla_{NDM}$, $bla_{VIM}$, $mcr-1$, $mecA$, and $vanA$ displayed more variability in their distribution. Notably, $bla_{NDM}$ and $bla_{VIM}$ concentrations exhibited a broad distribution, ranging from undetected to $3 \times 10^{-5}$ and $8 \times 10^{-4}$ copies per 16S rRNA gene, respectively. All targets in this group showed either non-detects (i.e. all wastewater samples collected during the week of sampling were non-detects for the WWTP) or partial detects (i.e., at least one sample collected for the WWTP during week of sampling was a non-detect). $Mcr-1$ had the highest number of non-detects (38.3% of sites), and partial detects (32% of sites), followed by $vanA$ (24.5% of sites non-detects and 22.7% of sites partial detects) and $bla_{NDM}$ (21.5% of sites non-detects and 16.6% of sites partial detects). Overall, most 16S rRNA normalized ARGs in wastewater were positively correlated with each other (Spearman's correlation coefficients ranged from 0.17 to 0.88, all $p < 0.05$), with beta-lactamase genes showing some of the strongest pairwise correlations, except for $bla_{VIM}$ (Fig. S1A).

To summarize the resistance burden, we calculated z-scores based on the distributions of each gene normalized by 16S rRNA gene concentrations, generating burden scores for each resistance class and total resistance. Although more than the presence of ARGs is needed to estimate burden, we will use this term here to mean antibiotic resistance gene prevalence in the community's wastewater. Burden z-scores were highly correlated (Fig. S1B). Comparison of these burden scores across census regions revealed significant regional difference in total resistance burden (Kruskal-Wallis test, $p < 0.05$; Fig. 1C). Specifically, the South had a significantly higher total resistance burden compared to the Midwest. Additionally, the West and the South demonstrated significantly higher colistin resistance burdens than the Midwest and the Northeast (Fig. 1D; Post-hoc Conover-Iman test, $p < 0.05$, Benjamini-Hochberg correction applied). For individual beta-lactamase genes, significant regional differences were observed without a clear trend (Fig. 1D). For instance, the Midwest had significantly higher $bla_{CTX-M}$ levels but lower $bla_{KPC}$ concentrations (Fig. S2). Regional differences among all ARGs can be seen in Fig. S2. The two genes with the highest national variability, $bla_{NDM}$ and $bla_{VIM}$, displayed distinct patterns: $bla_{NDM}$ showed significant regional differences, with the South and the West having higher concentrations than the Midwest, while $bla_{VIM}$ exhibited similar variability across all regions.

### Determinants of wastewater antibiotic resistance burden

Using publicly available datasets, we investigated the influence of specific anthropogenic or socioeconomic factors on the antibiotic resistance burden observed in wastewater. Values for secondary variables for sites were determined at either the sewershed scale or for the county predominantly intersecting a sewershed (see Supporting Information). From our correlational analysis between resistance gene prevalence and potential determinants, several notable associations were identified. Colistin resistance measured in wastewater was associated with overcrowding, speaking limited English, and certain race and ethnicities ($p < 0.05$ after Benjamini-Hochberg adjustment; Fig. 2A). Overall beta-lactamase resistance was associated weakly with many variables but specific beta-lactamase genes ($bla_{CMY}$, $bla_{KPC}$, $bla_{NDM}$, $bla_{TEM}$) showed stronger correlations with socioeconomic factors such as being uninsured, being under housing burden, overcrowding, and lack of a high school diploma, as well as speaking limited English and race and ethnicity variables ($p < 0.05$ after Benjamini-Hochberg adjustment; Fig. 2B). Among these four genes, $bla_{NDM}$ exhibited the greatest number of significant correlations with socioeconomic factors. Antibiotic prescriptions from Epic Cosmos and the number of animals had little to no correlation with the resistance

burden. These correlations were adjusted with Benjamini–Hochberg method to control the false discovery rate (Fig. 2); unadjusted correlation coefficients are provided in Fig. S3.

Total antibiotic resistance burden score differed significantly based on the presence of hospitals and whether the number of nursing homes in the sewershed exceeded the median (4) (Fig. 2C). While colistin resistance gene concentration normalized by 16S rRNA did not differ based on the presence of hospitals or nursing homes, higher *mcr-1* concentrations were associated with the presence of airports, higher population density, and urbanicity (Fig. 2D). When examining individual beta-lactamase genes, some genes, such as $bla_{CMY}$ showed no significant difference with presence of points of interest. In contrast, other genes like $bla_{NDM}$, $bla_{KPC}$, $bla_{TEM}$, and $bla_{VIM}$ showed significant differences based on presence of hospitals and the number of nursing homes (Figs. 2E, F; S4E, H, I). Notably, higher $bla_{NDM}$ concentration was the only beta-lactamase gene that was associated with the presence of airports, higher population density, and urbanicity (Fig. 2F).

### Predictors of antimicrobial resistance concentrations

Using publicly available datasets of non-clinical determinants of antibiotic resistance and antibiotic prescription rates obtained from Epic Cosmos, we developed random forest models to predict wastewater ARG concentrations across U.S. counties with characteristics similar to our sampled wastewater sewersheds. The random forest models presented here had one tuning parameter: the number of variables randomly sampled at each split (node), which ranged from 2 to 28. For all ARGs, the model that minimized RMSE considered 2 variables at each split, so all subsequent results are presented with this configuration. Overall, the predictive performance for individual genes, indicated by $R^2$, ranged from 0.14 to 0.46 (Fig. S5). The *mcr-1* model, showing colistin resistance, achieved one of the highest predictive performances with an $R^2$ of 0.46, indicating it could explain 46% of the variance in colistin resistance across counties based solely on the secondary dataset. The models for $bla_{KPC}$ and $bla_{NDM}$, both conferring carbapenem resistance and classified as urgent threats by the CDC, showed similar predictive performance with $R^2$ values of 0.30 and 0.37, respectively. The *vanA* model, showing resistance to vancomycin, had an $R^2$ of 0.34 (Fig. 3). Variables identified to be the most important across all individual ARG prediction models were proportion of the population being under housing burden, being uninsured, certain race and ethnicities, speaking limited English, and urbanicity of the sewershed (Fig. 4).

### Discussion

This study establishes a quantitative baseline for the current concentrations of ARGs in wastewater across the U.S. and provides new insights for understanding the complex factors driving resistance gene abundance circulating at the population level. We present evidence that the antibiotic resistance burden disproportionately affects low-income and disadvantaged communities. Increased levels of ARGs in wastewater were associated with limited access to health insurance, housing burden, and overcrowding. Health insurance access and housing burden were also among the most frequently identified socioeconomic variables of importance in our random forest prediction model. In particular, colistin resistance and certain beta-lactamase genes ($bla_{CMY}$, $bla_{KPC}$, $bla_{NDM}$, and $bla_{TEM}$) exhibited some of the strongest positive associations with these socioeconomic factors. It is important to consider that some of these indicators are co-correlated as shown in our PCA (Fig. S7), reflecting complex, interwoven societal challenges. Therefore, the associations we observe may not be attributable to any single vulnerability factor in isolation but rather to the broader context of socioeconomic disadvantage or specific community characteristics that these variables collectively represent. These findings align with previous studies linking socioeconomic status to

risk of AMR, based on clinical isolates, sewage metagenomes, and sewage cultured antibiotic resistant bacteria[6,7,19–21]. For example, in a recent study, Goetgeluck et al. cultured bacteria from 12 diverse sewersheds in Atlanta and found that sewersheds with higher proportions of Hispanic, non-Hispanic Asian, non-English speaking residents, and crowded households exhibited higher concentrations of fluoroquinolone-resistant Enterobacterales and third-generation cephalosporin-resistant bacteria (*E. coli*, *Klebsiella*, *Enterobacter*, or *Citrobacter* spp.), similar to significant potential determinants found in our study[22]. Cooper et al. found significant spatial correlation in Dallas-Fort Worth, Texas, metropolitan area between AMR organisms (particularly AmpC beta-lactamase and methicillin-resistant *Staphylococcus aureus*), and high levels of the Area Deprivation Index, which includes factors like income, education, employment, and housing quality[19]. In our study, $bla_{CMY}$, an AmpC beta-lactamase gene, was associated with being uninsured, overcrowding, and low education levels. Race was also significantly and positively associated with many of the resistance genes, and minority race variables were the most frequently identified important variables across all potential determinants in the random forest model. Race can be correlated with systemic disadvantages, such as limited access to healthcare and education, which are well-documented risk factors for resistant infections[20,23]. For instance, Black communities and areas with high rates of poverty often experience higher rates of MRSA infections[20,21]. Overall, these findings underscore that the risk of antibiotic resistance infections is linked to socioeconomic disparities. While our study reveals compelling links between various societal factors and ARG concentrations, these findings are correlations and do not imply causation.

We also identified international travel and recent immigration as potential drivers of increased ARG burden in domestic wastewater. $bla_{NDM}$ and colistin resistance (*mcr-1*) were significantly associated with the presence of airports and the proportion of the population who speak limited English. We interpret limited English proficiency as a potential proxy for populations with stronger ties to, and more frequent travel to, their countries of origin, which often coincides with recent immigration. This association between AMR and populations who speak limited English was also identified in a recent study examining antibiotic resistant bacteria from multiple sewersheds with varying socioeconomic statuses in Atlanta[22], in addition to previous research showing that international travelers frequently acquire AMR organisms, contributing to the global spread of resistance genes[8,9,24,25]. Depending on travel destination and behavior, the acquisition rate of extended-spectrum beta-lactamase-producing Enterobacterales has been shown to be as high as 75%[24]. Another study by Sridhar et al. also found *mcr*-mediated colistin-resistant Enterobacterales in metagenomes from post-travel stool samples, with the highest associations found among travelers to South America and South-Eastern Asia[25]. Here, *mcr-1* also showed significant associations with the presence of Hispanic/Latino and Asian populations. In the U.S., high cost of healthcare and lack of insurance in low-income populations can be a crucial reason for international travel. For instance, a survey of adults in Los Angeles by Macias and Morales found that many low-income and uninsured individuals seek low-cost health care and medications in Mexico to meet their urgent health care needs, despite the significant burden of travel and cost[26]. This also ties international travel to similar socioeconomic factors identified in our study that correlate with wastewater ARG concentrations. In addition, non-prescription use of antibiotics may be influenced by cultural differences among recent immigrants. In a cross-sectional survey conducted by Corbett et al. in Colorado, significant gaps were found in antibiotic resistance awareness between English and non-English speakers, with ethnicity serving as a predictor of knowledge, attitudes, and awareness regarding antibiotic use and resistance[27]. Similarly, Hawkes et al. identified race to be a major factor related to distrust in the healthcare system, leading to

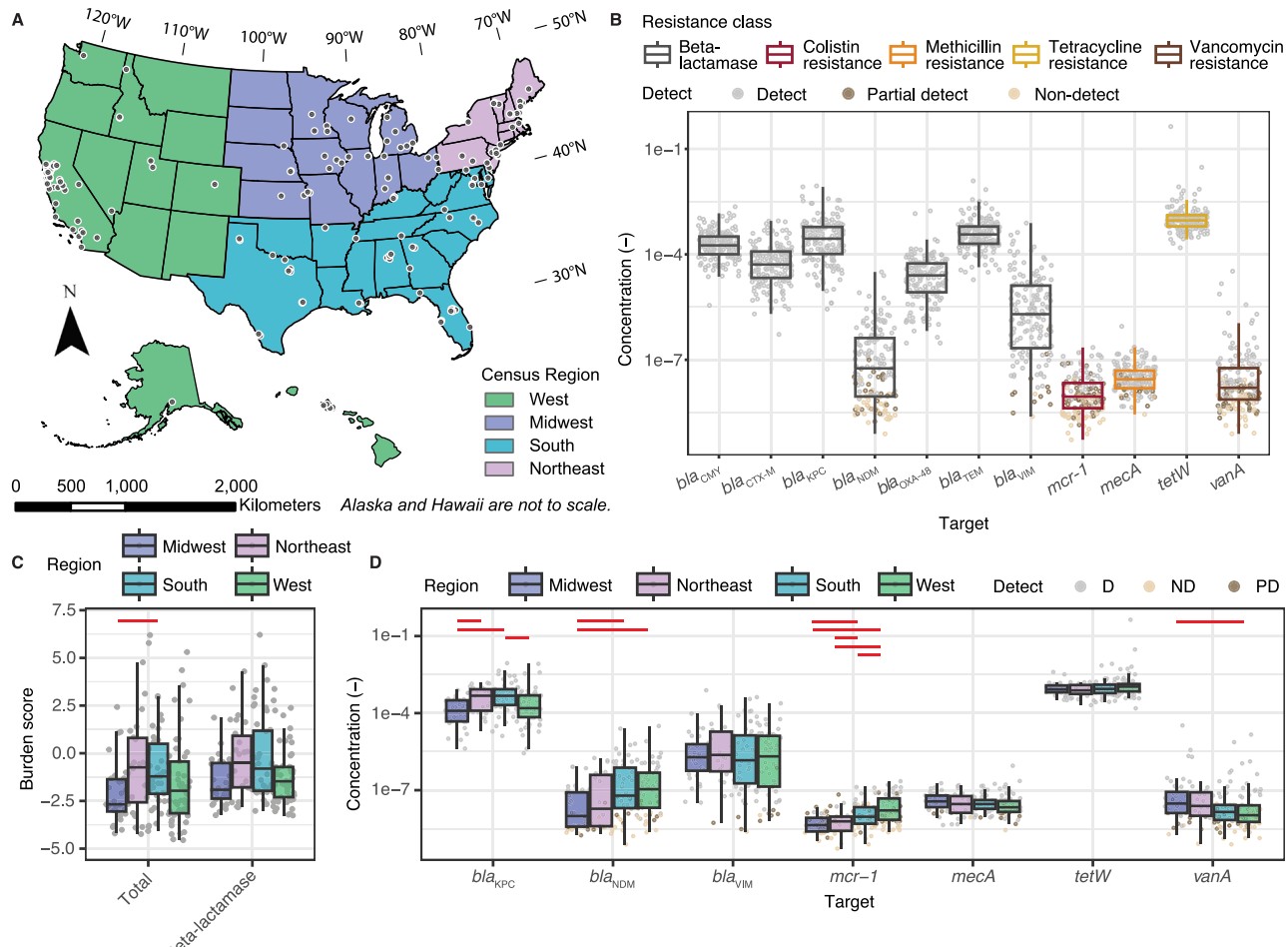

**Fig. 1 | Overall and regional distribution of antibiotic resistance genes (ARGs) and antibiotic resistance burden. A** Sample locations (black circles) for this study divided into census regions. Created in ArcGIS Pro (version 3.1.1) using 2023 state cartographic boundaries (20 m resolution) from the US Census Bureau (accessed August 5, 2024)[64]. **B** ARG concentration normalized by 16S rRNA gene for all sample sites ($n = 163$ for all except *mcr-1* where $n = 162$). **C** Total resistance burden and beta-lactamase resistance burden calculated as z-scores for each census region ($n = 32$ for Midwest, $n = 24$ for Northeast, $n = 48$ for South, and $n = 59$ for West). **D** Select beta-lactamase genes and colistin, methicillin, tetracycline, and vancomycin resistance genes in concentrations normalized by 16S rRNA gene across census regions

($n = 32$ for Midwest, $n = 24$ for Northeast, $n = 48$ for South, and $n = 59$ for West). Complete list of the ARGs can be seen in Fig. S2. Each symbol in (**B**–**D**) represents a wastewater treatment plant. If the gene was undetected, half of the theoretical measurement limit was substituted as measured value. The median is shown by the line inside the box with the 25th and 75th percentile represented by the lower and upper boundary of the box. Bottom and top whiskers show 1.5× interquartile range. Red lines indicate a significant pairwise difference between regions measured by the two-tail Conover-Iman post-hoc test with Benjamini-Hochberg correction applied ($p < 0.025$; exact *p*-values provided in Table S2).

non-prescription antibiotic use[28]. However, there is some evidence suggesting that Americans of international heritage may have different attitudes and behaviors towards antibiotic use than more recent immigrants[29]. Therefore, the correlation between race and ARG prevalence in wastewater may be driven by a combination of socioeconomic factors and immigration status, emphasizing that societal and cultural patterns influence antibiotic use.

Antibiotic use itself showed weak or no correlation with presence of ARGs. This lack of correlation may be due to discrepancies between the amount of prescribed antibiotics in healthcare settings and actual antibiotic consumption across humans and animals. Riquelme et al. found that global trends in sewage antibiotic concentrations, measured with LC-MS/MS, did not align with trends predicted from reported human antibiotic sales data[12]. While this discrepancy may be less pronounced within the U.S., it underscores a disconnect between antibiotic sales data and actual consumption, which likely reflects large antibiotic usage in food and companion animals that is also expected to drive antibiotic resistance[30]. Another potential factor contributing to this disconnect could be non-prescription antibiotic use, which is not captured by prescription data. Although the overall outpatient

antibiotics use in the U.S. has gradually declined (potentially indicating more careful prescribing from clinicians)[31], greater attention must be directed toward antibiotic use in animals and non-prescription usage, both of which cannot be addressed through hospital-based antibiotic stewardship programs. Depending on the population characteristics, non-prescription antibiotic use has been shown to vary from 1% to 66%[32]. Moreover, the comparative variable importance ranking of antibiotic prescription rate and other non-clinical indicators in our random forest model reinforces our finding that antibiotic resistance is driven by broader social, environmental, and infrastructural factors beyond just prescribed antibiotic use. These findings suggest that current antibiotic stewardship efforts may be insufficient in addressing the broader societal dynamics that fuel the global antibiotic resistance crisis.

By examining the quantitative levels of individual ARGs across the nation, we observed varying degrees of national variation, which informs the utility of certain genes as indicators of broader resistance patterns. With our extensive sampling across 163 wastewater treatment plants, we confirmed that several high-concentration targets, including *bla*CMY, *bla*CTX-M, *bla*KPC, *bla*TEM, and *tetW*, exhibit broad

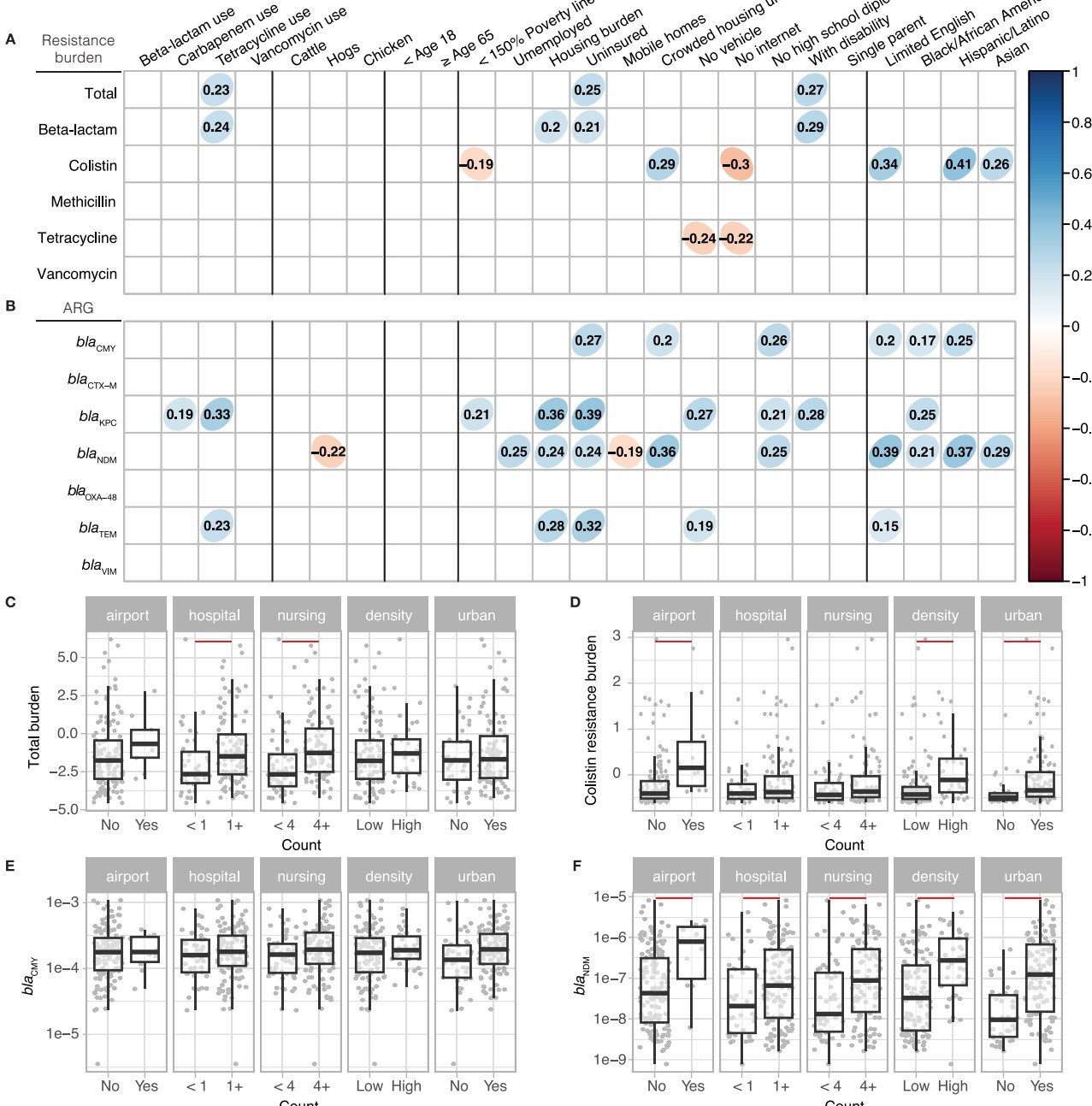

**Fig. 2 | Association between antibiotic resistance found in wastewater and potential determinants.** One-sided Spearman's correlation coefficient among potential determinants of resistance burden and **A** antibiotic resistance burden score and **B** beta-lactamase gene concentrations measured in wastewater and normalized by 16S rRNA gene ($p < 0.05$ with Benjamin-Hochberg correction applied). Bivariate analysis of points of interest, population density, and urbancity for **C** total antibiotic resistance burden and **D** colistin resistance burden measured as z-scores and **E** $bla_{CMY}$ and **F** $bla_{NDM}$ concentration measured in wastewater normalized by 16S rRNA. For **C**–**F**, $n = 153$ for <1 and $n = 10$ for 1+ airport, $n = 44$ for <1 and $n = 119$ for 1+ hospital, $n = 59$ and $n = 104$ for 4+ nursing homes, $n = 38$ for high and $n = 125$ for low population density, and n = 44 for rural and $n = 119$ for urban areas. The median is shown by the line inside the box with the 25th and 75th percentile represented by the lower and upper boundary of the box. Bottom and top whiskers show 1.5× interquartile range. Red lines indicate a significant pairwise difference between regions measured by the one-sided Wilcoxon post-hoc test with Benjamini-Hochberg correction applied ($p < 0.05$). Specific statistics of Wilcoxon post-hoc test, including exact $p$-values, are shown in Table S3. Bivariate analysis for antibiotic resistance genes not shown here, can be seen in Fig. S4.

ubiquity across centralized wastewater systems in the U.S. Out of these high-concentration targets, genes like *tetW* that exhibit minimal variation nationwide may not offer much insight into the resistance burden since its distribution is relatively consistent. This high prevalence and general abundance of tetracycline resistance genes is consistent with patterns previously reported in wastewater studies[33,34]. On the other hand, genes with larger variation, such as $bla_{NDM}$ and $bla_{VIM}$, may provide more meaningful information about resistance dynamics.

Notably, across individual genes, we observed significant differences in regional variation and correlation with potential determinants. The maps shown in Fig. 3 reinforce regional differences in antibiotic resistance gene concentrations, and the variable importance ranking (Fig. 4) gives potential insight for these differences. The use of z-scores to aggregate resistance burden allowed for a standardized comparison, but may obscure gene-specific insights that were obtained when examining individual genes, consistent with other summary metrics

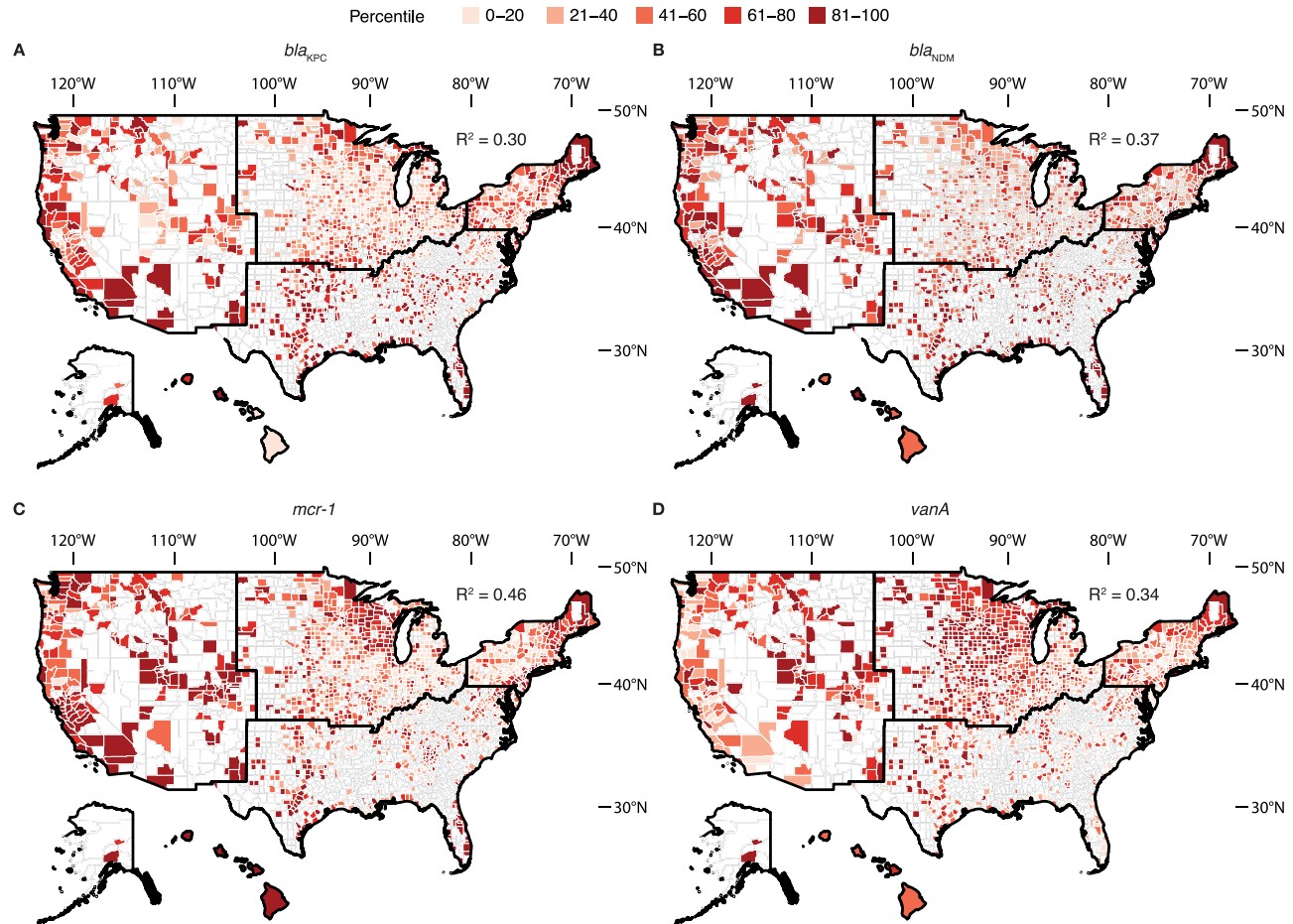

**Fig. 3 | Random forest modeling of antibiotic resistance gene concentrations across the United States using secondary data.** Predicted **A** $bla_{KPC}$ ($R^2$: 0.30), **B** $bla_{NDM}$ ($R^2$: 0.37), **C** $mcr-1$ ($R^2$: 0.46), and **D** $vanA$ ($R^2$: 0.34) concentrations across the United States. The predicted concentration, normalized by 16S rRNA, is visualized as percentiles, with darker colors indicating higher concentrations. Random forest models are trained using the secondary data in Table 1. Counties that were not predicted to avoid extrapolating outside our training set are colored white. Random forest modeling results for antibiotic resistance genes not shown are in Fig. S5. Alaska and Hawaii are not to scale.

used in previous studies[11,35]. This highlights the value of measuring individual genes and modeling their abundance separately, and underscores the difficulty of developing an overall resistance burden indicator. The random forest model also performed less effectively with z-scores representing overall resistance (Fig. S8). Consequently, while useful for a high-level overview, the aggregated burden map should be interpreted with caution, as it may not fully capture the nuanced heterogeneity observed in the underlying gene-specific data or the localized observations from our sampling sites. This aligned with the correlation analysis where more insights were gained by examining individual genes, underscoring a need for granular analysis to understand community-level resistance dynamics.

This study has several limitations. First, while the study covers 40 out of 50 states of the U.S. and includes 163 sites, regional coverage of wastewater samples received was uneven, with some areas, such as California, overrepresented. Additionally, matching sewershed boundaries to counties or census tracts may have introduced inaccuracies when linking resistance data to local characteristics, especially in training our random forest prediction models. Second, it is important to note that this is a cross-sectional analysis and while this design provides a valuable national baseline of ARG concentrations, it does not permit assessment of temporal or seasonal influences. The literature on ARG seasonality in wastewater is complex; while some studies report no obvious trends in time and only a modest effect of seasonality[11], others have identified significant seasonal patterns[11,36-39].

As such, researchers intending to use this dataset as a baseline for future studies should note that this is a cross-sectional data set that does not capture temporal (seasonal) variability of ARGs. Additionally, if any factors lead to a delayed response in the wastewater signal, as recent studies suggest could occur with antibiotic use[40,41], this study would not be able to assess such effects. Third, our use of ddPCR does not reveal the genomic context of the ARGs detected; however, all the targets in this study are either plasmid-mediated resistance genes or can be carried on mobile genetic elements, which increases their risk of transmission into human microbiota. Our analysis is generalizable to the U.S. and potentially other high-income settings, but other studies in low and middle income countries would likely identify different risk factors[7]. Lastly, the $R^2$ values of the random forest model suggest that the chosen set of determinants explains a substantial portion of the variation in ARG concentrations, but there are other unmeasured factors influencing resistance patterns across the country that were not captured in this study. For example, the size of hospitals and nursing homes may also influence wastewater ARG levels, but we could not weight facilities by size due to lack of metadata in the hospital and nursing home datasets. Similarly, weather effects were not considered, although it is possible that severe rainwater events could affect the concentrations of ARGs either due to infiltration or a dilution effect to a small proportion ($n = 21$) of our treatment plants that had combined sewer systems (systems where stormwater and sewer water are combined).

| $bla_{CMY}$ | $bla_{CTX-M}$ | $bla_{KPC}$ | $bla_{NDM}$ | $bla_{OXA-48}$ | $bla_{TEM}$ | $bla_{VIM}$ | $mcr-1$ | $mecA$ | $tetW$ | $vanA$ | |
|---|---|---|---|---|---|---|---|---|---|---|---|
| 57 | 89 | 100 | 100 | 47 | 78 | 42 | 29 | 54 | 36 | 22 | Housing burden |
| 66 | 100 | 91 | 36 | 41 | 100 | 36 | 27 | 57 | 44 | 29 | Uninsured |
| 44 | 97 | 100 | 32 | 10 | 66 | 100 | 33 | 64 | 56 | 23 | Urban |
| 48 | 94 | 47 | 21 | 100 | 47 | 82 | 40 | 66 | 47 | 19 | Asian |
| 55 | 62 | 43 | 41 | 23 | 70 | 80 | 100 | 64 | 50 | 19 | Limited English |
| 49 | 46 | 64 | 37 | 30 | 76 | 76 | 31 | 63 | 100 | 35 | Population Density |
| 85 | 76 | 62 | 60 | 7 | 95 | 40 | 24 | 65 | 48 | 26 | Black/African American |
| 52 | 64 | 65 | 25 | 10 | 61 | 75 | 66 | 69 | 68 | 30 | Hispanic/Latino |
| 55 | 56 | 54 | 25 | 11 | 83 | 30 | 70 | 73 | 56 | 35 | No high school diploma |
| 65 | 69 | 39 | 21 | 39 | 46 | 65 | 52 | 59 | 44 | 29 | No internet |
| 68 | 84 | 48 | 40 | 34 | 43 | 32 | 30 | 49 | 43 | 34 | < 150% Poverty line |
| 59 | 56 | 29 | 19 | 8 | 33 | 100 | 22 | 72 | 53 | 55 | Hogs |
| 43 | 65 | 43 | 71 | 12 | 36 | 78 | 25 | 74 | 34 | 17 | Mobile homes |
| 33 | 68 | 52 | 39 | 22 | 39 | 21 | 16 | 66 | 42 | 100 | Tetracycline usage |
| 100 | 67 | 49 | 14 | 17 | 31 | 30 | 21 | 100 | 41 | 24 | Chicken |
| 36 | 51 | 60 | 15 | 19 | 61 | 25 | 31 | 74 | 96 | 25 | Beta-lactam usage |
| 69 | 66 | 52 | 35 | 11 | 42 | 46 | 28 | 55 | 51 | 38 | Cattle |
| 47 | 54 | 30 | 25 | 30 | 35 | 36 | 86 | 59 | 59 | 28 | Crowded housing units |
| 57 | 62 | 38 | 55 | 10 | 30 | 36 | 35 | 71 | 65 | 24 | Vancomycin usage |
| 63 | 74 | 46 | 22 | 20 | 32 | 38 | 29 | 64 | 62 | 29 | With disability |
| 33 | 73 | 52 | 22 | 19 | 64 | 29 | 37 | 63 | 47 | 24 | No vehicle |
| 45 | 58 | 49 | 52 | 6 | 52 | 29 | 30 | 63 | 38 | 34 | < Age 18 |
| 44 | 45 | 43 | 47 | 7 | 46 | 67 | 20 | 63 | 46 | 24 | ≥ Age 65 |
| 25 | 38 | 48 | 33 | 9 | 58 | 30 | 35 | 48 | 70 | 57 | Nursing |
| 50 | 56 | 44 | 21 | 11 | 48 | 34 | 26 | 78 | 45 | 37 | Single parent |
| 51 | 67 | 39 | 35 | 10 | 45 | 30 | 26 | 63 | 41 | 30 | Unemployed |
| 42 | 47 | 54 | 25 | 8 | 42 | 28 | 22 | 73 | 55 | 27 | Carbapenem usage |
| 23 | 31 | 53 | 47 | 10 | 36 | 25 | 42 | 43 | 59 | 33 | Hospitals |

| 0 | 20 | 40 | 60 | 80 | 100 |

**Fig. 4 | Summary of variable importance for random forest prediction models for each antibiotic resistance gene (ARG).** The rows represent different variables used in the random forest models, sorted by their overall importance, summed across all genes, from highest to lowest. The color gradient reflects the magnitude of the relative importance of each variable for each ARG prediction model, with red indicating higher importance. Individual prediction model importance plots are shown in Fig. S6.

This study presents a quantitative analysis of clinically relevant ARGs in wastewater, offering a valuable tool and baseline for understanding and monitoring resistance dynamics across the U.S. The use of wastewater to study ARGs allows for real-time, population-level insights into the presence and distribution of antibiotic resistance, supplementing clinical data that may be underestimating the resistance burden. The robust baseline established by this study can be used to track changes in resistance gene prevalence over time, which could help identify emerging resistance trends, track the impact of interventions, and better predict the trajectory of resistance in the population. One of the significant merits of this study is its sensitivity in detecting rare but clinically important resistance genes enabled by the use of digital PCR. This method allows for the identification of low-abundance resistance genes that can be missed in studies relying on non-targeted sequencing methods like metagenomics[16,18]. This enhanced sensitivity is critical for accurate detection of emerging resistance genes, such as colistin resistance or certain beta-lactamase genes that are low abundance, which can have profound implications for clinical treatment strategies. Our findings provide important insights into the complex relationship between socioeconomic factors and antibiotic resistance, suggesting that antibiotic stewardship is not

sufficient on its own but improving access to healthcare and raising the standard of living in the U.S. is necessary to reduce antibiotic resistance.

## Methods
The authors confirm that this research complies with all relevant ethical regulations. Wastewater sampling and analysis did not involve human participants or the collection of identifiable private information; therefore, institutional review board oversight was not required for this component of the study. Access to antibiotic prescription data used in this study was obtained through the Epic Cosmos platform. In accordance with the Epic Cosmos data use and publication policies, the study protocol followed the established approval process, which included formal notification of the intent to publish and the submission of the manuscript for review and feedback prior to submission.

### ddPCR assay optimization
We chose probe-based qPCR assays in literature that have been utilized for detection of the target ARGs in wastewater to test on the ddPCR platform (Supplementary Data 1). The assays were initially tested on gblocks (IDT; Supplementary Data 1) of the target region of each ARG

as either a singleplex (FAM) or a duplex (HEX) assay with an annealing temperature gradient ranging from 55 °C to 65 °C in order to identify an annealing temperature that yielded sufficient separation between negative and positive droplets (Fig. S9). Subsequently, the assays were tested using templates at four different dilutions (neat, diluted in molecular grade water at 1:10, 1:100, 1:1000) of wastewater samples collected from seven wastewater treatment plants in the U.S. with distinct geographical characteristics to identify the dilution factor suitable for each target ARG. Consequently, we categorized the target ARGs into two groups based on their expected concentrations in wastewater (high versus low) in order to multiplex the assays into two 6-plex assay mixes that utilized an annealing temperature of 58 °C.

## Sample collection

Wastewater treatment plant (WWTP) staff provided either "grab" samples from the primary clarifier or 24-hour composite influent samples from the headworks during a one-week period in May 2024 (Supplementary Data 2 shows which WWTP collected each type and the date range of the collected samples). The "grab" samples in this case are solids collected in the primary clarifier over 1–8 h, representing composite community wastewater solids. The samples were then stored at 4 °C, shipped to the laboratory overnight, and processed within 48 h of receipt at the laboratory. A total of 443 samples were collected and analyzed for this study (Fig. 1A, Table S1). The total population served by the treatment plants was 24,455,284, representing 7% of the U.S. population.

## Pre-analytical processing and nucleic-acid extraction

Our study focused on the analysis of wastewater solids for ARG quantification. This decision was informed by previous comparative studies, including our own work, which have demonstrated that ARGs are reliably detectable and highly correlated across different wastewater matrices, including raw influent, centrifuged influent pellets, and wastewater solids. Specifically, our prior research showed comparable detection sensitivity and strong correlation in ARG concentration between these sample types, with wastewater solids exhibiting slightly higher ARG concentrations on a per-mass basis[42]. This indicates that wastewater solids serve as a representative and effective matrix for assessing community-level ARG prevalence and that centrifuged influent pellets were a suitable substitute when wastewater solids were not available.

Methods for pre-analytical processing and nucleic-acid extraction and purification have been described in detail by Boehm et al., so are briefly summarized here[43]. Wastewater solids were obtained from the sample by use of Imhof cones for influent samples and centrifugation for primary settled solids and then suspended in a buffer (DNA/RNA shield, Zymo Research, R1100-250, California, USA) containing bovine coronavirus (as a positive extraction control; Calf-Guard Cattle Vaccine, Ohio, USA) at a concentration known to reduce the potential for inhibition in the analytical steps[44]. The samples were homogenized after addition of grinding balls (5/32-in., OPS Diagnostics, New Jersey, USA) and then nucleic acids extracted and purified from 6 to 10 aliquots of the supernatant to obtain 6 to 10 300 µl replicate extracts from each sample (see Supplementary Data 2 for whether 6 or 10 replicates used for the plants). Nucleic acid extraction and inhibitor removal were conducted using Chemagic Viral DNA/RNA 300 kit H96 (Revvity, CMG-1033-S, Massachusetts, USA) for the Chemagic 360 and Zymo OneStep-96 PCR Inhibitor Removal kit (Zymo Research, D6030, California, USA). The methods for nucleic acid extraction are described step-by-step in protocols.io[45]. The first two aliquots of every sample were chosen to be analyzed for ARGs. Extraction-negative controls (water) were extracted using the same protocol as the homogenized samples. Nucleic acids were used immediately for quantification of ARGs. A portion of the solids was dried to determine their dry weight[43].

## Droplet generation and quantification

The probe used for each assay was labeled with one of six fluorophores (FAM, HEX, Cy5, Cy5.5, ROX, and/or ATTO590) and run in a 6-plex assay using the probe-mixing approach (Supplementary Data 1). All targets were multiplexed and run together based on being high concentration (samples diluted 1:100 with ultrapure $H_2O$) or low concentration (undiluted samples) in wastewater. Two wells were run per reaction using the first two nucleic acid extractions. Digital droplet PCR was performed on 20 µL samples from a 22 µL reaction volume, which consisted of a 5.5 µL template. Final primer concentrations were 900 nM and probe concentrations were 250 nM for all ARG targets. All ARG master mixes were prepared using ddPCR Supermix for Probes (BioRad #1863010, California, USA). Droplet generation was performed using an Automated Droplet Generator (BioRad #1864101, California, USA). Quantification was performed using a QX600 Droplet Digital PCR system (BioRad #17007769, California, USA). All assays were performed at an annealing temperature of 58 °C. Thermocycling conditions were as follows: 5 min at 95 °C, followed by 40 cycles of 30 s at 95 °C and 1 min at 58 °C, followed by 10 min at 98 °C, and an infinite hold at 4 °C. Each plate run included no template controls, extraction negative controls, extraction positive controls, and positive controls, which all performed as expected. Extraction negative controls used ultra-pure water, extraction positive controls used BCoV spiked into each wastewater sample, and positive controls were synthetic IDT gblocks, which included the target amplicon. No template controls and extraction negative controls were negative for all targets in all runs. Extraction positive controls and template positive controls were positive in all runs. For each well in ddPCR, a minimum of 10,000 droplets was required, and at least three positive droplets were required to be quantifiable. If a target was undetected from a diluted sample, the measurement was redone or excluded (one *mecA*, six *bla*$_{OXA-48}$, and one *mcr-1* measurement excluded). In cases of duplicate measurements at different dilutions, the value with the smaller ddPCR confidence interval was selected. Two replicate wells were merged per measurement. Thresholding for each target was performed automatically using the QX Manager Software Version 2.2 (BioRad, #168032, California, USA).

After quantification of ARGs, the leftover nucleic acid replicates were combined and stored at −80 °C for 17 days to 21 days prior to being used as templates for measuring high copy number bacterial target, 16S rRNA. 16S rRNA genes were quantified using ddPCR Eva-Green Supermix (BioRad, #1864034, California, USA) and QX200 Droplet Digital PCR system (BioRad, #1864001, California, USA). Eva-Green is an intercalating dye that binds to double stranded DNA. Final primer concentrations were 100 nM for 22 µL reaction volume, 5.5 µL of which was the template. 16S rRNA quantification was run in single-plex reactions using template diluted 1:50,000 with UltraPure $H_2O$. One well was run per reaction using the combined template from the quantification of ARG genes. Thermocycling conditions were as follows: 5 min at 95 °C, followed by 40 cycles of 30 s at 95 °C and 1 min at 55 °C, followed by 5 min at 4 °C, 5 min at 90 °C and infinite hold at 4 °C. Each plate run included no template controls (NTCs). The NTCs resulted in a small number of positive droplets (average 16 positive droplets) and therefore any sample concentration on the same plate with less than an order of magnitude difference was excluded from analysis (three 16S rRNA measurements excluded). All wells had more than 10,000 partitions. Thresholding for each target was performed automatically using QX Manager Software Version 2.2.

Additional details associated with the Minimal Information MIQE Experiments (dMIQE) for digital droplet PCR reporting are described here[46]. The average (standard deviation) number of droplets for the quantification of ARGs was 34,225 (7143; 2 merged wells), and for bacterial count was 17,997 (1443; 1 well). The volume of partitions, as reported by BioRad, was 0.000795 µL. The mean and standard deviation of copies per partition for each target is shown in Table S4. BCoV

**Table 1 | Demographic and socioeconomic variables that may influence antibiotic resistance**

| Sewershed Characteristic | Description |
|---|---|
| <150% Poverty line | Proportion of the population for whom poverty status is determined below the 150% poverty line in the sewershed area |
| Unemployed | Proportion of the civilian labor force age 16+ years that is unemployed in the sewershed area |
| Housing burden | Proportion of occupied housing units that are housing cost-burdened with an annual income <$75 K in the sewershed area |
| No high school diploma | Proportion the population age 25+ years with no high school diploma in the sewershed area |
| Uninsured | Proportion of the civilian noninstitutionalized population that is uninsured in the sewershed area |
| ≥ Age 65 | Proportion of the population age 65+ years in the sewershed area |
| <Age 18 | Proportion of the population age 17 years and younger in the sewershed area |
| With disability | Proportion of the civilian noninstitutionalized population with a disability in the sewershed area |
| Single parent | Proportion of single-parent households with children <18 years in the sewershed area |
| Limited English | Proportion of persons age 5+ years who speak English "less than well" in the sewershed area |
| Black/African American | Proportion of the population that is Black/African American, not Hispanic or Latino in the sewershed area |
| Hispanic/Latino | Proportion of the population this is Hispanic or Latino in the sewershed area |
| Asian | Proportion of the population that is Asian, not Hispanic or Latino in the sewershed area |
| Mobile homes | Proportion of housing units that are mobile homes in the sewershed area |
| Crowded housing units | Proportion of occupied housing units with more people than rooms in the sewershed area |
| No vehicle | Proportion of occupied housing units with no vehicle in the sewershed area |
| No internet | Proportion of households with no internet in the sewershed area |

Refer to the Supporting Information for further information about how values for each characteristic were determined for the sewersheds included in this study.

served as an extraction and inhibition control to assess nucleic acid recovery efficiency and to identify samples with significant PCR inhibition. Median bovine coronavirus (BCoV) recoveries (BCoV measured in sample / BCoV measured in DNA/RNA shield without sample) were, on average 1.06 (25th percentile = 0.71, 75th percentile = 1.3) across all samples. Recovery higher than 100% is likely due to an underestimation of the amount of BCoV spiked into the samples. Based on the recovery, we concluded that there was no gross inhibition of the samples, and no samples were removed due to low BCoV recovery. After QA/QC, we had 443 different samples and 5309 measurements that represented 163 WWTPs in 40 states and 134 cities in the U.S.

### Demographic, socioeconomic, and environmental data

Antibiotic resistance may be influenced by various demographic, socioeconomic, and environmental determinants—in addition to antibiotic usage. Demographic and socioeconomic factors affect a community's capability to respond and adapt to external hazards and stressors, such as public health emergencies. A community's status as urban or rural or its population density may affect its access to and quality of healthcare services. Proximity to major points of entry (e.g., airports) and health care facilities (e.g., hospitals, nursing homes) may facilitate the importation or transmission of public health threats in communities. Antibiotics are also commonly used for agricultural purposes, and ARGs may be transmitted zoonotically within communities. Therefore, we characterized sewersheds based on several demographic, socioeconomic (Table 1), environmental, and antibiotic usage determinants (Table 2) to assess their relationship with wastewater concentrations of ARGs.

Most WWTPs (n = 94 of 163) provided a sewershed boundary. For WWTPs that did not provide a sewershed boundary (n = 69 of 163), we approximated a boundary based on the zip code(s) serviced by the WWTP and 2023 USA ZIP Code Boundaries published by Esri (Source: TomTom, US Postal Service, Esri)[47]. Because sewershed boundaries do not align with traditional boundaries (e.g., counties or census tracts), we approximated sewershed-level estimates for each determinant in Tables 1 and 2 as described in the Supporting Information. The demographic and socioeconomic determinants we selected from the American Community Survey (ACS) are based on the 2022 Social Vulnerability Index (SVI) from the US CDC, as social vulnerability describes the degree to which demographic and socioeconomic

factors affect a community's resilience[48,49]. We derived urbanicity and population density from the US Census Bureau and ACS, respectively[49,50]. Sewersheds with an urban area proportion ≥ 50% were classified as urban, and sewersheds with an urban area proportion <50% were classified as non-urban for data analysis. We obtained locations of major airports from Esri (source: Federal Aviation Administration's National Airspace System Resource Aeronautical Data Product) and locations of hospitals and nursing homes from the US Department of Homeland Security's Homeland Infrastructure Foundation-Level Data database[51,52]. We obtained animal inventory numbers from the US Department of Agriculture National Agricultural Statistics Service[53]. Antibiotic prescription was determined using Epic Cosmos (Epic Cosmos, Epic Systems Corporation, Wisconsin), a dataset created in collaboration with a community of Epic health systems representing more than 284 million patient records from over 1,500 hospitals and 36,000 clinics from all 50 states, D.C., Lebanon, and Saudi Arabia, though only U.S. data was used in this study[54,55]. The Epic Cosmos data includes all healthcare systems which use the Epic electronic healthcare record system. This has been shown to be nationally representative from a demographics perspective and nationally representative of tested sub-populations[55,56]. To calculate antibiotic usage, all encounters—which represent any visit, outpatient or inpatient, between a patient and healthcare provider—were selected for every county available in Epic Cosmos between June 1, 2023, and June 1, 2024. From the selected encounters, the number of prescriptions for each antibiotic class was calculated for each county, and the total number of encounters in the county was used to normalize the data. A total of 1,590,238,036 patient encounters were analyzed.

### Data analysis

Analysis was conducted using R (Version 4.3.0), ArcGIS Pro (version 3.1.1), and the following packages: caret (version 7.0-1), dplyr (version 1.1.2), EnvStats (version 2.8.1), corrplot (version 0.92), conover.test (version 1.1.6), stats (version 4.3.0), ggplot2 (version 3.5.1), and pheatmap (version 1.0.12). Outlier detection was performed using Rosner's test on 16S rRNA to exclude significant outliers. One 16S rRNA measurement was excluded. Each time point was treated as a replicate for each wastewater treatment plant and therefore averaged to obtain a final representative concentration of each target. This approach was specifically designed to mitigate the influence of short-term

**Table 2 | Environmental and antibiotic prescription variables that may influence antibiotic resistance**

| Sewershed Characteristic | Description |
|---|---|
| Urban | Proportion of the sewershed area that is classified as urban |
| Density | Population density (people per square kilometer) of the county predominantly intersecting the sewershed |
| Airport | Number of major airports in the sewershed area |
| Hospital | Number of hospitals in the sewershed area |
| Nursing | Number of nursing homes in the sewershed area |
| Cattle | Number of cattle in the county predominantly intersecting the sewershed |
| Chicken | Number of chickens in the county predominantly intersecting the sewershed |
| Hogs | Number of hogs in the county predominantly intersecting the sewershed |
| Beta-lactam use | Proportion of encounters with beta-lactam antibiotics (penicillins and cephalosporins) dispensed in the county predominantly intersecting the sewershed |
| Carbapenem use | Proportion of encounters with carbapenem antibiotics dispensed in the county predominantly intersecting the sewershed |
| Tetracycline use | Proportion of encounters with tetracycline antibiotics dispensed in the county predominantly intersecting the sewershed |
| Vancomycin use | Proportion of encounters with vancomycin antibiotics dispensed in the county predominantly intersecting the sewershed |

Refer to the Supporting Information for further information about how values for each characteristic were determined for the sewersheds included in this study.

fluctuations in wastewater characteristics, thereby providing a pragmatic method for a more robust and representative average concentration for each site during the cross-sectional sampling period. Samples were categorized as non-detect (ND) if none of the replicates detected the target, detect (D) if all replicates detected it, and partial detect (PD) if at least one of the replicates did not detect the target. For non-detect values, half of the lower measurement limit was calculated and assigned as a substitute value[57]. Summary of the analysis when 0 was substituted for non-detect values is shown in Fig. S10. All ARG concentrations were normalized by 16S rRNA concentration to account for variations in bacterial biomass in each sample[58,59]. Z-scores were calculated for each target by comparing the concentration to the overall distribution across all samples, using the mean ($\mu$) and standard deviation ($\sigma$) in the following equation: $z = \frac{x-\mu}{\sigma}$. The AMR burden was calculated by summing up the z-scores of all ARG targets. The beta-lactamase burden was computed by summing up the z-scores of $bla_{CMY}$, $bla_{CTX-M}$, $bla_{KPC}$, $bla_{NDM}$, $bla_{OXA-48}$, $bla_{TEM}$, and $bla_{VIM}$. Data points with z-scores greater than 3 or less than -3 were excluded from analysis involving secondary datasets (four $bla_{CMY}$, five $bla_{CTX-M}$, three $bla_{KPC}$, two $bla_{NDM}$, two $bla_{OXA-48}$, two $bla_{TEM}$, three $bla_{VIM}$, six $mcr-1$, one $mecA$, one $tetW$, and two $vanA$ measurements excluded).

For statistical analysis, the Shapiro-Wilk test was used to assess normality, and Spearman's correlation coefficient was employed to evaluate relationships between variables, as the data were not normally distributed. For determinants of resistance with insufficient distribution (number of airports, number of hospitals, number of nursing homes, population density, and urbanicity), the data were split into two groups to conduct a bivariate analysis using the Wilcoxon rank sum test to determine if there was a significant difference in wastewater ARG concentrations between the lower and upper values of potential determinants. Number of airports and hospitals was split into groups based on the presence of the facilities, and the number of nursing homes was split by the national median number of nursing homes (median: 4). The sewersheds were considered urban if there was 50% or more area considered to be urban. Population density was split by 617 per km², which is the average number of people in cities in the U.S. The number of animals was analyzed as percentile ranking from the national dataset. Principal component analysis was conducted on determinants of resistance to visualize covariance among the secondary variables. The Kruskal-Wallis test was applied to assess differences across Census regions, followed by the Conover-Iman test for post-hoc analysis to identify significant pairwise differences. The Benjamini-Hochberg correction was applied when potential determinants were involved and when investigating regional differences to control for the false discovery rate.

## Predictive modeling

Based on our secondary data's characteristics, we chose random forest modeling to predict wastewater ARG concentrations in counties in the United States. Random forest modeling allows us to use data with varied distributions and nonlinear relationships, is resistant to outliers, provides interpretable feature importance, and is robust to covariance between independent variables[60,61]. Given our limited sample size for the cross-sectional analysis, the random forest model minimizes the risk of overfitting often associated with more complex architectures (such as neural networks and gradient boost models) when trained on small datasets. We first aggregated our mean wastewater ARG measurements at the county level. If a county was represented more than once in the dataset (i.e. contained 2 treatment plants), one of the sets of measurements from that county was randomly selected to represent the entire county. We chose this approach instead of averaging the values to minimize structural bias, as averaging would systematically and incorrectly assign equal weight to all plants regardless of their population size, which we could not correct for since wastewater treatment plants typically do not report the number of population served broken down by county boundaries. A sensitivity analysis regarding this approach can be found in the SI. This left 108 unique observations. All variables from Table 1, excluding number of airports, were considered and obtained for each county in the U.S. Airport data was omitted from this analysis due to a highly skewed distribution in which almost all included sewersheds had no airports (93% of training data locations, 97.5% of nationwide locations). The variables for number of nursing homes, number of hospitals, and number of animals were turned into a percentile rank before analysis. To train the models, we used the caret package (version 7.0-1) in R[62]. To avoid extrapolating extensively to rural or unsewered areas, we limited the predictive model to counties that have secondary data values within our minimum and maximum ranges of the training data set for all variables. For example, if our training dataset included 0–1000 nursing homes, only counties that had 0–1000 nursing homes were included. Ten-fold cross-validation was used to evaluate the models, and we varied the number of features at each split between 2 and 28. Models were selected based on minimizing root mean square error. Variable importance was assessed using the varImp() function from the caret package, which randomly permutes variables and assesses the impact on prediction accuracy (Fig. S6). More details on our modeling approach can be found in the SI.

## Reporting summary

Further information on research design is available in the Nature Portfolio Reporting Summary linked to this article.

## Data availability

Data generated in this study on antibiotic resistance gene concentrations in 163 wastewater treatment plants across the United States have been deposited in the Stanford Data Repository under accession code vb318cm9509 (https://purl.stanford.edu/vb318cm9509). The antibiotic prescription data from Epic Cosmos are available under restricted access due to data privacy laws. These data cannot be redistributed or hosted in public repositories to maintain proprietary data protections. Access can be obtained through affiliation and approval by a Cosmos participating organization or by contacting Epic Cosmos at https://cosmos.epic.com/request-access/. Other secondary data used in this study are publicly available from data sources mentioned in the Methods section.

## Code availability

All analysis R code used for this manuscript (https://doi.org/10.5281/ZENODO.18749049) is available on GitHub: https://github.com/sooyeolkim/ww_arg [63].

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

## Acknowledgements

We thank the participating wastewater treatment plants for their samples for the project. This work was supported by a gift to A.B.B. from the Sergey Brin Family Foundation. R.Y.L. was supported by the National Institutes of Health (NIH) Institutional National Research Service Award T32 AI007502. A.J.P. is a Chan Zuckerberg San Francisco Biohub Investigator.

## Author contributions

A.J.P. and A.B.B. conceptualized the project. S.K., A.Z., E.M.G.C., D.D., R.Y.L., and C.M. acquired primary and secondary data in the study. S.K., A.Z., and E.M.G.C. performed the analysis. A.J.P., A.B.B., M.K.W., and B.J.W. supervised the project. S.K., A.Z., and E.M.G.C. wrote the first draft. All authors reviewed and edited the final manuscript.

## Competing interests

The authors declare no competing interests.
