## [Peer Review file · Nature Communications]

Wastewater surveillance reveals patterns of antibiotic resistance across the United States

Corresponding Author: Professor Amy Pickering

Version 0:

Reviewer comments:

Reviewer #1

(Remarks to the Author)

The contribution by Kim et al. has many strengths, but also some considerable weaknesses. The strength of the work lies in their extensive dataset of ARG concentrations from sites located across the US. The authors are clearly able to show that there is substantial variability in the concentrations of ARGs in different regions of the US. This is highly useful information that can help inform our collective understanding of ARG dissemination. The weaknesses of the work result primarily from the authors' attempt to not only consider the local sewershed-scale determinants of AMR, but to then extend their results nationally. As noted in my general comments #2-4 (below) there are many questions raised by their approaches to define the variables included in their assessments of the local importance of the different determinants. The US-scale simulations of ARG concentrations that are central to this work are fully dependent upon the quality of the data used to develop their random forest models and as such that data and its interpretation must be fully transparent.

At the present time, there appears to be too much extrapolation of what they have learned from their 163 sites (which are geographically skewed towards a few specific regions (primarily California)) to the much broader US. Relatedly, a further challenge to the potential validity of their US assessment is that their dataset is restricted to regions served by sewer networks; however, substantial portions of the US are unsewered and as such the relative importance of such unsewered sites (which often serve lower income regions) cannot be estimated using sewer-based data alone. These potential limitations (particularly given the relatively low R^2 values reported for their models) require further consideration. One potential approach might be for the authors to more geographically limit the modeling to specific regions characterized by large numbers of sampling sites.

General Comments:

1. Much of the work described in this contribution is reliant upon the development of correlations between measured ARG levels and metadata obtained from outside sources. At the end of the day, even if the correlations are strong, they remain correlations and cannot be used to infer causation. Throughout the manuscript the authors need to include statements that reflect this fact. Further, when discussing these correlations the manuscript needs to better incorporate consideration of co-correlation in cases where the different metadata parameters are subject to significant covariance.
2. Additional information needs to be provided to better explain many of the variables included in Table 1. How were the numbers of airports, hospitals, and nursing homes considered? Was there a local regional area considered or do they have to lie within the expected sewershed? It seems straightforward to consider the hospitals and nursing homes within the geographical area that overlays the sewershed; however, the number of airports would seem to be less easily defined since a sewershed may not overlap with an airport even though the people within the sewershed heavily use a nearby airport. Further, how is the size of the airport, hospital, or nursing home taken into account? One would anticipate that larger facilities might have a much more substantial impact on ARG levels detected in sewage.
3. Based upon the data presented in Figures S12-S13 it appears that a majority of the sewersheds being considered contain no airports, no hospitals, no nursing homes, and no cattle, chickens, or hogs. Further, in the case of the three agricultural animals only a limited number of sites contain either chickens or hogs, but with substantial variation in total numbers (i.e., between 10^7 and 0). Given the relative paucity of sewersheds with any of these variables it seems very unlikely that their evaluation of their role as 'determinants of wastewater antibiotic resistance burden' [Lines 158-186] can be accurately

assessed and then accounted for in their subsequent US model development. Nonetheless, the authors attempt to do so as indicated by their text on lines 178-180: '...higher mcr-1 concentrations were associated with the presence of airports...' and lines 269-271 'We also identified international travel and recent immigration...as significant drivers of increased ARG burden. NDM and colistin resistance were significantly associated with the number of airports...' [As an aside - how did the authors determine that international travel and immigration were important? Solely based on nearby airports? If so, that would seem to be quite an intellectual leap.] Based upon Figure S12 only ~5 sites had any airports at all. It would seem more appropriate to not consider these variables when looking at potential associations with ARG levels (or total burdens). Similarly, by extension, the US modeling should be re-evaluated without consideration of these variables.

4. One of the potential key contributions of this manuscript is the modeled prediction of ARG prevalence across all counties in the U.S. However, it is not clear exactly how this model was constructed given the limited size of the initial dataset (163 wastewater sites with a majority present within California). As the authors illustrate throughout the manuscript, the factors that drive the dissemination of antibiotic resistance are highly complex, often quite inter-related, and site-specific. Accordingly, it seems to be quite the intellectual leap to think that their model is able to accurately predict ARG levels for regions of the country that are not included within their model training set. As illustrated in Figures S10 and S11 the distributions are heavily skewed towards urban areas that are only moderately populated – thus, the capability of their model to properly predict what is happening in either rural (potentially unsewered) or dense-urban cities would seem to be low.

5. One of the key sources for metadata is the Epic Cosmos database. Unfortunately, the authors description of how this database was used is insufficient. Epic Cosmos is a semi-closed resource that not everyone (including this reviewer) has access to and thus it is unclear what types of data it provides. The SI makes use of the term 'encounter', but this term is not defined. Is it a drug prescription? Or something else?

Additional Specific Comments:

1. Line 27. Is it necessary to make the statement starting with 'This is an underestimate...'? Such statements may belong in the main body of the paper, but it is not clear it belongs in the abstract which should focus on the results produced as part of the work.
2. Lines 66-67. Include commas after 'clinical isolates' and 'antimicrobial resistance levels'.
3. Lines 78-79. The use of the word 'most' '...most research to date has focused on the fate of ARGs across the wastewater treatment processes' is incorrect. There is a large body of research that has reported ARG levels in raw sewage over many years. Similarly, the statement 'have mostly been conducted using shotgun metagenomics' is also incorrect. There have been numerous reports wherein PCR-based methods have been used to quantify ARGs in raw sewage.
4. Line 85. The authors suggest that metagenomics cannot detect 'rare targets' but provide no literature justification of that point. The typical assumption is that PCR provides lower levels of detection, but since it is a targeted approach it is unable to detect untargeted analytes. I suggest rewriting to highlight the better detection limit for PCR-based methods and not make any claims about 'rare targets' – particularly since such a use of PCR is not revisited later in the manuscript.
5. Lines 101-123. It would be useful to compare the results presented in this manuscript to those obtained previously. How do the ARG levels compare? Should the reader expect that CMY, CTX-M, KPC, OXA-48, TEM, and tetW are ubiquitous? Should tetW be present at the highest level?
6. Line 125. Not all readers may be familiar with 'z-scores' and thus it would be useful either here or in the Materials and Methods to mathematically define how they are calculated.
7. Lines 176-186. How is urbanicity defined? The SI considers the distribution of urbanicity along a scale from 0 to 100, but Figure 2 defines urbanicity using a Yes/No scale. Is >50% urban a Yes or is something closer to 90% considered urban? The clear definition needs to be provided. A similar lack of clarity also affects the authors use of the term 'urban' – Figure 2 defines it using a Yes/No scale, but what that means in the context of Figure S11 is unclear. How dense does an area need to be to be considered urban?

Reviewer #2

(Remarks to the Author)

Thanks for asking me to review this manuscript. It was a pleasure to read.

In brief, Pickering et al., sought to understand factors that may contribute (associate with) to the distribution of AMR (in particular 11 targeted ARGs) across the US by measuring their abundance at 163 separate WWTP in the US. These sites included those within 40 separate states. They integrated this data with publicly available data on antimicrobial use and census data to look in a holistic fashion on factors that associate with AMR.

The manuscript is particularly well written and easy to follow. The use of the same colour for different regions made the manuscript easy to read. Its broad scope and relevance make it an ideal candidate for consideration of publication in Nature Communications. Perhaps the most contentious (only in some circles) is the finding that local antimicrobial prescribing (in this regulated, controlled environment) only have a modest impact. This is really important and a bit more discussion on this (what is there is great) would be fantastic given the magnitude of this global crisis.

There are several issues that should be addressed to make the manuscript more robust and easier to follow.

Major issues.

1. Details on the sewersheds that are being monitored and when they are being assessed are key – and are lacking here.
 - a. Please include a table on the nature of each of the WWTP - and the size of the population served. As the authors rightfully highlight the comprehensive nature of their study (i.e. 40 states included!) - it would be nice to see what fraction of the US

population is accounted for in these samples which would allow the reader to determine how much stock to put into their projections across each county.

- b. What are the influences of rainwater in the system? Some communities have separate storm and sewage wastewater systems - and although many do not. While 16S will somewhat mitigate this - it should be commented on.
 - i. The authors use the term grab samples – but also describe this as potentially being an aggregate sample collected over 1-8. Please distinguish if this was the same or different – and include in the table relative to those that provided 24-hours composite samples. Are differences observed between them based on diurnal patterns of water usage?
- c. Samples appear to have been connected at a single point (or week in time) - are these the same through the entire study? If not - please indicate when for each site. What are there seasonal influences?
 - i. While authors have argued some studies have not shown seasonal influence of AMR distribution – many, many have particularly as it pertains to respiratory viral seasons influence
 - ii. Limited serial monitoring with only 2.7 samples per WWTP all collected (I think) within one week. The authors do not describe or include information as to whether this was done at the same time and what time this might be. While some m
- d. Social variables in these WWTP require more details.
 - i. Airports - what was defined as an airport? Are these only large hubs - by some specific definition or does it include even small non-commercial strips? While definitions report # of hospitals the author's analysis is presented as presence/absence.
 - ii. Hospitals/LTC - how are hospitals defined? Are they only tertiary or quaternary sites or specifically large LTC facilities? I find it hard to imagine that any of these large catchments for WWTP don't include a small community hospital or nursing home? While these are presented as < or > 7 – what does this mean? 1 large nursing home of 350 individuals is likely more relevant than 8 small nursing homes of 40 individuals....
 - iii. More detail on all of these variables would help the reader.
2. MecA is not - as stated - a beta-lactamase and should not be grouped with others. If the authors wish to continue to include it – they will need to modify several figures. Its inclusion, however, in this list as an AMR threat is a bit misleading. MecA is relevant in so far as its presence in *S. aureus* is a threat, however, the vast majority of its measured abundance will derive from CONS which would not be considered an AMR threat.
3. Quality control is important – but how relevant is supplemented BCoV - an RNA target well established for SARS-CoV-2 - as a control in an AMR experiment? No results are presented on this – and I suggest to remove it entirely as it does not seem to be relevant at all. If you think so – please describe why it is important and provide data demonstrating which samples were excluded owing to the results generated.
4. I am unfamiliar about EPIC Cosmos and its ability to provide comprehensive antimicrobial usage data. Is this human only - or does it include animals and agriculture? Does it include hospitals or just communities? Is it represent ALL antimicrobial prescribing or just a small sampling? - please provide some background even if it is in the supplement for the reader to understand. Currently supporting citation is the company.
 - a. From this data the authors have assessed beta-lactam usage. However, lumping all beta-lactam antibiotics together for assessing selection influence for AMR is likely inappropriate. Beta-lactams can be very narrow- or very broad- spectrum. What the authors have done is like lumping all things with wheels together Some with wheels might be unicycles others semi-trailer trucks and others yet are 747s.... It is not the wheel (i.e. the beta-lactam ring) that differentiates them. The authors should at the very least look at narrow (e.g. penicillin, amoxicillin, ampicillin) vs broad (amoxi-clav, amp-clav, pip-tzp, ert, mem, imi) etc separately.
5. The authors have identified the highest burden of AMR is in the NE and the S – and this is clear in Figure 1. However, figure S7 looking at overall AMR and S6 for specific targets suggest that within the NE there is considerable heterogeneity with much of this small area having a low burden of resistance. Please comment on those factors contributing -and the heterogeneity that is apparent in the data.
6. The authors substitution of not-detected as half of the limit of detection should be cited – although this is well supported in other chemical wastewater based studies. I do fear however, that this may have influenced some of their analyses.
 - a. As mcr was identified the least (i.e. negative in 38.3% samples) - how much significance are we seeing in the population modelling as a result of their assumption of not detected being equivalent of half the limit of detection? What if these were changed to zero? Given its strongest associations with SES factors this would be wise. This is also true for NDM (21.5% non-detects).
7. The authors do not (to the degree required) – distinguish between association and causation in their discussion. This would be very important particularly as it pertains to the SES factors assessed here – and some caveats through the manuscript would be wise.
 - a. A multivariate analysis - particularly with such a large dataset to try to disentangle the many observed associations to determine which are independent would be nice.

Minor issues.

1. The authors do a good job in the figures – but less so in the text in emphasizing that their data has been normalized for 16S rRNA.
 - a. They do not also describe the rationale for this step
- b. Please ensure consistency in naming. 16S, 16s rRNA
2. The authors bounce around between ddPCR and dPCR. Were both used – and if so where and when? Or was only one methodology employed.
3. There are inconsistencies using proper formatting for genes vs proteins throughout the figures and texts.
4. I think a spelling mistake here “At each split” - or each site? Line 205.
5. Why are the data not always presented for each of the ARG-targets in each of the figure and analyses? (ie Fig S4 only 7 of the 11 gene targets (many beta-lactamases but not all) - and these differ from Figure S4 where a different 7/11 are presented . Figure S3 has 8/11. Figure S6 11/11.
 - a. If there is a rationale for the presenting only these selective examples please provide that.

Reviewer #3

(Remarks to the Author)

The authors measured 11 antibiotic resistance genes by digital droplet PCR in wastewater solids obtained from 163 wastewater treatment plants across the United States. The topic is very interesting, and the work has significant potential for impact. Further clarification on the following aspects is needed for readers to better understand its approach and potential impact.

Data Quality:

- Line 391: Wastewater concentrations vary significantly both diurnal and weekly. While the diurnal variation can be addressed using composite samples, the impact of weekly fluctuations remains unclear. Limited sampling (2.7 samples per plant) could introduce uncertainties. More details on the sampling and data processing are needed to justify the sufficiency of the average of 2.7 samples per wastewater treatment plant.
- Line 554: How different are the samples from multiple treatment plants within the same county? This is an important issue to examine before randomly keeping only one single value for these counties to build the RF models.
- It is not clear why only wastewater solids were analyzed and ARG in dissolved solids or the liquid fraction were excluded in this study. Can we assume the distribution of ARG in the liquid phase and suspended solids to be consistent and stable? Is ARG concentration in the wastewater solids representative of the total concentration in wastewater?

Effect of airport presence & international travel:

- How to interpret the effect of airport presence? A sewershed without an airport can either be a rural area or a large/mega city with the main airport just outside of the sewershed boundary. Many major cities have airports situated in neighboring municipalities or on the urban outskirts, rather than within the central urban area. This may explain why so many samples do not feature an airport (Figure 2).
- Line 271: This conclusion is not supported by the RF results, where the number of airports had a variable importance of zero, for mcr-1 and all other ARGs.

Model accuracy:

- Line 214: with R^2 values ranging from 0.30 to 0.40, can the explanation that a "significant amount of the underlying variation" is captured by non-clinical indicators be fully justified? Given the relatively low model performance, more details are needed to assess whether the RF models were trained properly to achieve the best possible accuracy. Key considerations include:
 - a. The quality of the ground truths: Were the wastewater samples representative of broader trends? More information on sampling strategy would help clarify this.
 - b. Input features selection and elimination: How were the 25 input feature selected? Was there a formal feature selection process? How to justify the appropriateness of the selected features?
 - c. Line 557: Why were antibiotic usage variables excluded from the RF models? Given their likely impact on ARG concentrations, how reliable are the variable importance results from a model that does not include these key predictors?
 - d. Line 204: why were the random forest models tuned using only one parameter, while there are many other parameters that can be tuned? Also, given the complex input-output relationships, how was the decision to use "only 2 variables at each split" justified?

Importance analysis:

- A major focus of this study is to understand the factors influencing ARG concentrations. This was explored through preliminary statistical analysis (Figure 2) and variable importance analysis using RF. Given the huge uncertainties with RF, why were alternative statistical/multivariate analysis techniques not considered?

Version 1:

Reviewer comments:

Reviewer #2

(Remarks to the Author)

I was pleased to re-review the manuscript by Pickering et al., I am pleased to see that it is again being considered for publication – it is an impactful manuscript that is broadly representative of the US population. The authors have done an excellent job of trying to address reviewer's comments (longer than the paper itself!) in a thorough and thoughtful manner. I am satisfied with these efforts – and the manuscript is both stronger, and easier to read. Most importantly the appropriate caveats and qualifiers are now included.

A few very, very, very minor comments;

Occasional areas in which nomenclature for genes is not adhered to (ie vanA in line 2019)

Lines 2610-263 comparing FQ resistant Enterobacterales vs 3rd gen cephalosporin resistant (E. Coli, Klebsiella, Enterobacter, Citrobacter ... these account for >90% of Enterobacterales clinical isolates... this could be rephrased to appear more consistent

I recommend this for FULL ACCEPTANCE – minor areas can be changed during the page setting step.

Reviewer #3

(Remarks to the Author)

I appreciate the authors' efforts to address the concerns raised during the initial round of review. However, several key issues have not been properly addressed. For example:

- Sampling limitations: I am unconvinced by the explanation regarding the limited sampling. While I acknowledge the logistical challenges associated with this large-scale study, the core issue lies in the representativeness of the snapshot data. Collecting samples within the same week does not sufficiently address this concern—wastewater characteristics are influenced by weather conditions and local water usage patterns, which can vary significantly across locations, even within the same week. A more robust justification for the representativeness of the snapshot data is required before the detailed results discussion can be considered acceptable. At a minimum, the limitations of the snapshot approach should be explicitly acknowledged and embedded throughout the discussion (though I don't think this alone would make the approach acceptable.)
- Random selection for RF modeling: The response regarding the decision to randomly keep a single value for counties with multiple treatment plants in the RF models remains insufficient. This decision needs to be justified by discussing other alternatives.
- Feature selection: The authors have modified their input feature set (e.g., removing airport presence, adding antibiotic usage variables), but there is no discussion on how these changes affected model performance or sensitivity. This, again, raises concerns about how the selected features are adequately justified.
- The factors influencing ARG concentrations: The decision to rely entirely on preliminary statistical analysis and RF variable importance for attribution analysis is not well-supported. There are many interpretability methods for models such as XGBoost and neural networks, and the statement that interpretability is lacking in these models is not valid and should be reconsidered. This links back to my earlier concern regarding the reliability of attribution analysis based on RF alone.

Reviewer #4

(Remarks to the Author)

The authors present a well constructed response to the initial reviewer feedback. I do not have any additional concerns. This is a very well written article that balances the nuances of WBS with the need to understand drivers in human communities.

Version 2:

Reviewer comments:

Reviewer #5

(Remarks to the Author)

I appreciate efforts made both by authors and other reviewers. I have provided my review comments (in red) as a mediating referee in the attachment.

(Remarks on code availability)

Below, please find the *original comments of the reviewers in black italic type*. Our responses are in indented plain text, and the changes within the body of the paper are provided as indented blue text.

Reviewer comments

Reviewer #1 (Remarks to the Author):

The contribution by Kim et al. has many strengths, but also some considerable weaknesses. The strength of the work lies in their extensive dataset of ARG concentrations from sites located across the US. The authors are clearly able to show that there is substantial variability in the concentrations of ARGs in different regions of the US. This is highly useful information that can help inform our collective understanding of ARG dissemination. The weaknesses of the work result primarily from the authors' attempt to not only consider the local sewershed-scale determinants of AMR, but to then extend their results nationally. As noted in my general comments #2-4 (below) there are many questions raised by their approaches to define the variables included in their assessments of the local importance of the different determinants. The US-scale simulations of ARG concentrations that are central to this work are fully dependent upon the quality of the data used to develop their random forest models and as such that data and its interpretation must be fully transparent.

At the present time, there appears to be too much extrapolation of what they have learned from their 163 sites (which are geographically skewed towards a few specific regions (primarily California)) to the much broader US. Relatedly, a further challenge to the potential validity of their US assessment is that their dataset is restricted to regions served by sewer networks; however, substantial portions of the US are unsewered and as such the relative importance of such unsewered sites (which often serve lower income regions) cannot be estimated using sewer-based data alone. These potential limitations (particularly given the relatively low R^2 values reported for their models) require further consideration. One potential approach might be for the authors to more geographically limit the modeling to specific regions characterized by large numbers of sampling sites.

We thank the reviewer for their thoughtful and constructive comments. We appreciate the detailed feedback regarding the limitations of our modeling approaches. We have addressed these concerns comprehensively, point-by-point, in our revised manuscript and response to reviewers below, including revising our predictive model to be limited to regions with similar characteristics to our training set, emphasizing the correlational nature of our analysis, and providing additional information on our secondary variables for improved clarity.

General Comments:

1. Much of the work described in this contribution is reliant upon the development of correlations between measured ARG levels and metadata obtained from outside sources. At the end of the day, even if the correlations are strong, they remain correlations and cannot be used to infer

causation. Throughout the manuscript the authors need to include statements that reflect this fact. Further, when discussing these correlations the manuscript needs to better incorporate consideration of co-correlation in cases where the different metadata parameters are subject to significant covariance.

We fully agree with the reviewer's point regarding the distinction between correlation and causation. Our study identifies associations between ARG levels and various metadata parameters, these relationships do not imply causation. Given the difficulties associated with conducting an experiment to vary the population characteristics we explore in the paper, we believe our paper is still a major contribution to investigating factors associated with AMR. We have carefully reviewed the manuscript and added explicit statements to clarify we are presenting correlations as follows:

While our study reveals compelling links between various societal factors and ARG concentrations, these findings are correlations and do not imply causation.

Additionally, we have edited the wording of our findings so that implications are framed in terms of "potential drivers" where appropriate to emphasize that the relationship is correlational and not causative.

We appreciate the reviewer's insight regarding co-correlation among the metadata parameters. We acknowledge that many of the parameters, especially the socioeconomic vulnerability indicators, are correlated. To show this, we have included in the SI a PCA analysis that shows where different variables cluster together (**Figure S7**). We recognize that this co-correlation complicates the interpretation of individual variable contributions. We have added text in the Discussion section to elaborate on how the interconnected nature of these variables means that observed binary associations might reflect broader underlying societal or environmental contexts rather than isolated effects of single parameters. Further, we chose random forest predictive modeling in part because it is robust to co-correlation between independent variables. The variables identified as important predictors in the random forest modeling are overall consistent with the bivariate associations presented. The following edits were made:

In the SI as Figure caption: A Principal Component Analysis (PCA) was performed on the metadata parameters to investigate the underlying structure and potential co-correlation. Secondary variables are categorized by color. The PCA indicated that the variance within the metadata was distributed across multiple dimensions, with the first two principal components explaining 36.7% of the total variance. Visual inspection of variable loadings revealed that while the social vulnerability indicators generally loaded positively on Dimension 1, suggesting a shared underlying characteristic, they displayed a considerable spread among Dimension 2. Subsequent components (PC3: 11.5%, PC4: 8.7%) captured further, distinct aspects of the data structure.

In the Discussion section: It is important to consider that some of these indicators are co-correlated as shown in our PCA (**Figure S7**), reflecting complex, interwoven societal challenges. Therefore, the associations we observe may not be attributable to any single

vulnerability factor in isolation but rather to the broader context of socioeconomic disadvantage or specific community characteristics that these variables collectively represent.

In the Materials and Methods section: Principal component analysis was conducted on determinants of resistance to visualize covariance among the secondary variables.

In the Materials and Methods section: Based on our secondary data's characteristics, we chose random forest modeling to predict wastewater ARG concentrations in every county in the United States. Random forest modeling allows us to use data with varied distributions and nonlinear relationships, is resistant to outliers, and provides interpretable feature importance, and is robust to covariance between independent variables.^{60,61}

2. Additional information needs to be provided to better explain many of the variables included in Table 1. How were the numbers of airports, hospitals, and nursing homes considered? Was there a local regional area considered or do they have to lie within the expected sewershed? It seems straightforward to consider the hospitals and nursing homes within the geographical area that overlays the sewershed; however, the number of airports would seem to be less easily defined since a sewershed may not overlap with an airport even though the people within the sewershed heavily use a nearby airport. Further, how is the size of the airport, hospital, or nursing home taken into account? One would anticipate that larger facilities might have a much more substantial impact on ARG levels detected in sewage.

As stated in the main text, complete data processing details for each variable in Table 1 are described in the SI (pp. 25-31). To address the reviewer's request for more explicit information about each variable in Table 1, we added the following details to the main text:

Most WWTPs (n = 94 of 163) provided a sewershed boundary. For WWTPs that did not provide a sewershed boundary (n = 69 of 163), we approximated a boundary based on the zip code(s) serviced by the WWTP and 2023 USA ZIP Code Boundaries published by Esri (Source: TomTom, US Postal Service, Esri).⁴⁹ Because sewershed boundaries do not align with traditional boundaries (e.g., counties or census tracts), we approximated sewershed-level estimates for each determinant in **Table 1** as described in the Supporting Information. The demographic and socioeconomic determinants we selected from the American Community Survey (ACS) are based on the 2022 Social Vulnerability Index (SVI) from the US CDC as social vulnerability describes the degree to which demographic and socioeconomic factors affect a community's resilience.^{50,51} We derived urbanicity and population density from the US Census Bureau and ACS, respectively.^{51,52} Sewersheds with an urban area proportion $\geq 50\%$ were classified as urban and sewersheds with an urban area proportion $< 50\%$ were classified as non-urban for data analysis. We obtained locations of major airports from Esri (source: Federal Aviation Administration's National Airspace System Resource Aeronautical Data

Product) and locations of hospitals and nursing homes from the US Department of Homeland Security’s Homeland Infrastructure Foundation-Level Data database.^{53,54} We obtained animal inventory numbers from the US Department of Agriculture National Agricultural Statistics Service.⁵⁵ Antibiotic usage was determined using Epic Cosmos (Epic Cosmos, Epic Systems Corporation, Wisconsin).^{56,57} The Epic Cosmos data includes all healthcare systems which use the Epic electronic healthcare record system. This has been shown to be nationally representative from a demographics perspective and nationally representative of tested sub-populations.⁵² To calculate antibiotic usage, all encounters—which represent any visit, outpatient or inpatient, between a patient and healthcare provider—were selected for every county available in Epic Cosmos between June 1, 2023 and June 1, 2024. From the selected encounters, the number of prescriptions for each antibiotic class was calculated for each county, and the total number of encounters in the county were used to normalize the data. A total of 1,590,238,036 patient encounters were analyzed.

Additionally, Table 1 and its footnote was edited to be as follows:

Table 1. Demographic, socioeconomic, environmental, and antibiotic usage variables that may influence antibiotic resistance

Sewershed Characteristic	Description
< 150% Poverty line	Proportion of the population for whom poverty status is determined below the 150% poverty line in the sewershed area
Unemployed	Proportion of the civilian labor force age 16+ years that is unemployed in the sewershed area
Housing burden	Proportion of occupied housing units that are housing cost-burdened with an annual income <\$75K in the sewershed area
No high school diploma	Proportion the population age 25+ years with no high school diploma in the sewershed area
Uninsured	Proportion of the civilian noninstitutionalized population that is uninsured in the sewershed area
≥ Age 65	Proportion of the population age 65+ years in the sewershed area
< Age 18	Proportion of the population age 17 years and younger in the sewershed area

With disability	Proportion of the civilian noninstitutionalized population with a disability in the sewershed area
Single parent	Proportion of single-parent households with children <18 years in the sewershed area
Limited English	Proportion of persons age 5+ years who speak English "less than well" in the sewershed area
Black/African American	Proportion of the population that is Black/African American, not Hispanic or Latino in the sewershed area
Hispanic/Latino	Proportion of the population this is Hispanic or Latino in the sewershed area
Asian	Proportion of the population that is Asian, not Hispanic or Latino in the sewershed area
Mobile homes	Proportion of housing units that are mobile homes in the sewershed area
Crowded housing units	Proportion of occupied housing units with more people than rooms in the sewershed area
No vehicle	Proportion of occupied housing units with no vehicle in the sewershed area
No internet	Proportion of households with no internet in the sewershed area
Urban	Proportion of the sewershed area that is classified as urban
Density	Population density (people per square kilometer) of the county predominantly intersecting the sewershed
Airport	Number of major airports in the sewershed area
Hospital	Number of hospitals in the sewershed area
Nursing	Number of nursing homes in the sewershed area
Cattle	Number of cattle in the county predominantly intersecting the sewershed

Chicken	Number of chickens in the county predominantly intersecting the sewershed
Hogs	Number of hogs in the county predominantly intersecting the sewershed
Beta-lactam use	Proportion of encounters with beta-lactam antibiotics (penicillins, cephalosporins, and carbapenems) dispensed in the county predominantly intersecting the sewershed
Tetracycline use	Proportion of encounters with tetracycline antibiotics dispensed in the county predominantly intersecting the sewershed
Vancomycin use	Proportion of encounters with vancomycin antibiotics dispensed in the county predominantly intersecting the sewershed

Refer to the Supporting Information for further information about how values for each characteristic were determined for the sewersheds included in this study.

As indicated in Table 1, the values for most variables, including airports, hospitals, and nursing homes, were determined using the geographical area that overlays the sewershed. Therefore, for airports, hospitals, and nursing homes, these points of interest had to intersect the sewershed area. We understand the reviewer’s concern about population mobility and airports; however, we chose to differentiate between sewersheds that directly receive wastewater input from one or more airports versus no airports. Population mobility is a limitation of all wastewater-based epidemiology work given that people may contribute to more than one sewershed throughout the day. For other variables (previously noted by footnote g and now explicitly noted in the “description” column of Table 1), the values were instead determined using the geographical area of the county that the sewershed predominantly overlays. The data processing steps used to determine the predominant county overlapping each sewershed are described in the SI. The geographical area of the county rather than the sewershed was used for these variables because the smallest spatial scale of these secondary datasets was the county scale.

As stated in the SI, the size of airports was taken into account by only including airports falling into the highest usage category as defined by the data source (Federal Aviation Administration’s National Airspace System Resource Aeronautical Data Product). Therefore, only major airports were considered in our analysis. For hospitals and nursing homes, we did not weight by size because not all sites in the data source (US Department of Homeland Security’s Homeland Infrastructure Foundation Level Data) reported a valid value for population or beds. We added the following to the SI and Discussion to address this limitation:

SI: We could not weight hospitals and nursing homes by size due to not all sites reporting a valid value for population or beds.

Discussion: The size of hospitals and nursing homes may also influence wastewater ARG levels, but we could not weight facilities by size due to lack of metadata in the hospital and nursing home datasets.

3. Based upon the data presented in Figures S12-S13 it appears that a majority of the sewersheds being considered contain no airports, no hospitals, no nursing homes, and no cattle, chickens, or hogs. Further, in the case of the three agricultural animals only a limited number of sites contain either chickens or hogs, but with substantial variation in total numbers (i.e., between 10^7 and 0). Given the relative paucity of sewersheds with any of these variables it seems very unlikely that their evaluation of their role as 'determinants of wastewater antibiotic resistance burden' [Lines 158-186] can be accurately assessed and then accounted for in their subsequent US model development. Nonetheless, the authors attempt to do so as indicated by their text on lines 178-180: '...higher *mcr-1* concentrations were associated with the presence of airports...' and lines 269-271 'We also identified international travel and recent immigration...as significant drivers of increased ARG burden. NDM and colistin resistance were significantly associated with the number of airports...' [As an aside - how did the authors determine that international travel and immigration were important? Solely based on nearby airports? If so, that would seem to be quite an intellectual leap.] Based upon Figure S12 only ~5 sites had any airports at all. It would seem more appropriate to not consider these variables when looking at potential associations with ARG levels (or total burdens). Similarly, by extension, the US modeling should be re-evaluated without consideration of these variables.

We appreciate the reviewer's attention to detail and want to clarify that the majority of sewersheds have 1 or more of each of these variables (except airports). The original SI figures did not reflect this due to the bin size of the histograms presented. We have recreated the SI figures as bar charts to distinctly show zero values as follows:

Figure S12, now S14:

Figure S13, now S15:

Our consideration of international travel and immigration was not based solely on the presence of nearby airports, but was conceptually driven by existing literature demonstrating the well-established role of global human movement in the dissemination of antibiotic resistance. Recent studies have highlighted that individuals who travel, especially to regions with higher AMR prevalence, can acquire new resistance genes during their travel. These studies are cited in the manuscript:

- Sridhar, S., Turbett, S. E., Harris, J. B. & LaRocque, R. C. Antimicrobial-resistant bacteria in international travelers. *Curr. Opin. Infect. Dis.* **34**, 423–431 (2021).
- Frost, I., Van Boeckel, T. P., Pires, J., Craig, J. & Laxminarayan, R. Global geographic trends in antimicrobial resistance: the role of international travel. *J. Travel Med.* **26**, taz036 (2019).
- Arcilla, M. S. *et al.* Import and spread of extended-spectrum β -lactamase-producing Enterobacteriaceae by international travellers (COMBAT study): a prospective, multicentre cohort study. *Lancet Infect. Dis.* **17**, 78–85 (2017).
- Sridhar, S. *et al.* Insights into global antimicrobial resistance dynamics through the sequencing of enteric bacteria from U.S. international travelers. Preprint at <https://doi.org/10.1101/2025.01.27.635056> (2025).

We acknowledge the reviewer’s observation that the proportion of airports in our sewersheds was low (6.1%, 10 out of 163 sites). Based on the reviewer’s suggestion we have removed airports from the random forest modeling. However, our decision to consider international travel and immigration as important factors is supported by two key pieces of evidence from our analysis:

1. When we performed statistical comparisons using Wilcoxon rank-sum tests between sewersheds with and without nearby airports, we observed significant differences in colistin resistance encoded by *mcr-1* concentrations.
2. Both our models and correlation analysis consistently identified limited English proficiency as a significant socioeconomic factor associated with higher ARG levels. We interpret this as a potential proxy for populations with stronger ties to, and more frequent travel to, their countries of origin. These individuals may have increased exposure to diverse resistomes through international travel or cultural practices, which aligns with the broader concept of immigration and global movement influencing local resistance patterns.

We have refined our discussion to articulate these points more clearly. The text reads:

We interpret limited English proficiency as a potential proxy for populations with stronger ties to, and more frequent travel to, their countries of origin, which often coincides with recent immigration. This association between AMR and populations who speak limited English was also identified in a recent preprint examining antibiotic resistant bacteria from multiple sewersheds with varying socioeconomic statuses in Atlanta,²² in addition to previous research showing that international travelers frequently acquire AMR organisms, contributing to the global spread of resistance genes.^{8,9,24,25}

Additionally, the text in Methods was updated to reflect the exclusion of airports from our RF model and the rationale behind the decision:

Airport data was omitted from this analysis due to a highly skewed distribution in which almost all included sewersheds had no airports (93% of training data locations, 97.5% of nationwide locations).

4. One of the potential key contributions of this manuscript is the modeled prediction of ARG prevalence across all counties in the U.S. However, it is not clear exactly how this model was constructed given the limited size of the initial dataset (163 wastewater sites with a majority present within California). As the authors illustrate throughout the manuscript, the factors that drive the dissemination of antibiotic resistance are highly complex, often quite inter-related, and site-specific. Accordingly, it seems to be quite the intellectual leap to think that their model is able to accurately predict ARG levels for regions of the country that are not included within their model training set. As illustrated in Figures S10 and S11 the distributions are heavily skewed towards urban areas that are only moderately populated – thus, the capability of their model to properly predict what is happening in either rural (potentially unsewered) or dense-urban cities would seem to be low.

We agree with the reviewer that our model may not be representative of rural and unsewered areas. As a way of addressing this, we have updated our modeling to only predict within counties that have secondary data values falling within our minimum and maximum ranges of the training data set. For example, if our training dataset included 0 to 1000 nursing homes, only counties that had between 0 to 1000 nursing homes will be predicted to avoid extrapolating extensively. We also note that the training data set includes several of the largest urban centers of the United States, including San Francisco and the greater Bay Area, Los Angeles and Los Angeles County, and Boston, as well as other major urban centers including Las Vegas, Dallas, Newark, Kansas City, Chicago, and Miami. As such, we believe the model is likely performing well at predicting these concentrations within both moderately populated and dense urban cities.

We have presented revised predictive maps in the paper and the SI. The excluded counties seem to align with unsewered areas of the U.S. All figures involving the RF model was updated and the Methods section was edited as follows to reflect this change:

To avoid extrapolating extensively to rural or unsewered areas, we limited the predictive model to counties that have secondary data values within our minimum and maximum ranges of the training data set for all variables. For example, if our training dataset included 0 to 1000 nursing homes, only counties that had 0 to 1000 nursing homes were included.

5. One of the key sources for metadata is the Epic Cosmos database. Unfortunately, the authors description of how this database was used is insufficient. Epic Cosmos is a semi-closed resource that not everyone (including this reviewer) has access to and thus it is unclear what types of data it provides. The SI makes use of the term ‘encounter’, but this term is not defined. Is it a drug prescription? Or something else?

We have added the following citations, which provide significantly more information on Epic Cosmos and how it can be used:

- Tarabichi, Y. *et al.* The Cosmos Collaborative: A Vendor-Facilitated Electronic Health Record Data Aggregation Platform. *ACI Open* **05**, e36–e46 (2021).
- Mankowski, M. A. *et al.* Generalizability of kidney transplant data in electronic health records — The Epic Cosmos database vs the Scientific Registry of Transplant Recipients. *Am. J. Transplant.* **25**, 744–755 (2025).

“Encounter” in this context is any clinical contact between patient and provider (telehealth included). To clarify this, we have also added text describing what an encounter is in the Methods as follows:

To calculate antibiotic usage, all encounters—which represent any visit, outpatient or inpatient, between a patient and healthcare provider—were selected for every county available in Epic Cosmos between June 1, 2023 and June 1, 2024. From the selected encounters, the number of prescriptions for each antibiotic class was calculated for each county, and the total number of encounters in the county were used to normalize the data.

Additional Specific Comments:

1. Line 27. *Is it necessary to make the statement starting with ‘This is an underestimate...’? Such statements may belong in the main body of the paper, but it is not clear it belongs in the abstract which should focus on the results produced as part of the work.*

The intent of the referred sentence in the comment was to briefly contextualize the public health burden and motivate the use of wastewater surveillance to capture non-clinical cases; however, we recognize that the abstract should prioritize direct findings and methods of the presented work. Therefore, we have removed this statement from the abstract and included it in the introduction instead to ensure that our abstract remains concise and fully focused on the study’s objectives, methods, and key results. The introduction now reads:

The 2019 Antibiotic Resistance Threats Report by the U.S. Centers for Disease Control and Prevention (CDC) highlights the severity of the crisis, reporting over 2.8 million antibiotic resistant infections annually, resulting in more than 35,000 deaths.² This is an underestimate, as it is based on people who seek medical attention.

2. Lines 66-67. *Include commas after ‘clinical isolates’ and ‘antimicrobial resistance levels’.*

We have added the commas in the mentioned locations for proper punctuation.

3. Lines 78-79. *The use of the word ‘most’ ‘...most research to date has focused on the fate of ARGs across the wastewater treatment processes’ is incorrect. There is a large body of research that has reported ARG levels in raw sewage over many years. Similarly, the statement ‘have mostly been conducted using shotgun metagenomics’ is also incorrect. There have been numerous reports wherein PCR-based methods have been used to quantify ARGs in raw sewage.*

We appreciate the reviewer's careful reading and suggested edits. We acknowledge that significant research has indeed quantified ARGs in raw sewage using PCR-based methods for various purposes although the goal of global and national antibiotic resistance surveillance is more recent. Our intent was to highlight the unique combination of national-scale community-level ARG surveillance in wastewater solids using highly sensitive dPCR. Therefore, we have revised these sentences in the introduction to more accurately reflect the existing landscape while still emphasizing what we believe is the contribution of this work. The introduction now reads:

While previous research has characterized ARG levels and their fate within wastewater treatment systems using PCR-based methods, studies undertaking national-scale community antibiotic resistance surveillance via wastewater monitoring are still emerging. Prior work assessing community-level ARGs in wastewater has largely involved either global spread or differences in the wastewater resistome globally,^{4,6,11,12} or are limited to single cities or states.^{13–15} Additionally, many previous wastewater ARG studies conducted for the purpose of monitoring community antibiotic resistance have used shotgun metagenomics...

4. Line 85. *The authors suggest that metagenomics cannot detect 'rare targets' but provide no literature justification of that point. The typical assumption is that PCR provides lower levels of detection, but since it is a targeted approach it is unable to detect untargeted analytes. I suggest rewriting to highlight the better detection limit for PCR-based methods and not make any claims about 'rare targets' – particularly since such a use of PCR is not revisited later in the manuscript.*

Based on this feedback, we have rewritten the text in the introduction to read: While metagenomic approaches are useful in understanding the overall ARG diversity, they often lack the sensitivity and quantitative precision needed for accurate abundance estimates. In contrast, digital PCR (dPCR) has the advantage of being highly sensitive and quantitative, allowing for normalized abundance estimates and comparison across samples in time and space.¹⁶⁻¹⁸

We have also moved the following citations that were originally in one of the discussion section paragraphs to the introduction to provide justification for the comparison of metagenomics to PCR-based methods:

- Maestre-Carballa, L., Navarro-López, V. & Martínez-García, M. City-scale monitoring of antibiotic resistance genes by digital PCR and metagenomics. *Environ. Microbiome* **19**, 16 (2024).
- Davis, B. C., Vikesland, P. J. & Pruden, A. Evaluating Quantitative Metagenomics for Environmental Monitoring of Antibiotic Resistance and Establishing Detection Limits. *Environ. Sci. Technol.* acs.est.4c08284 (2025) doi:10.1021/acs.est.4c08284.

- Knight, M. E. *et al.* National-scale antimicrobial resistance surveillance in wastewater: A comparative analysis of HT qPCR and metagenomic approaches. *Water Res.* **262**, 121989 (2024).

5. Lines 101-123. It would be useful to compare the results presented in this manuscript to those obtained previously. How do the ARG levels compare? Should the reader expect that CMY, CTX-M, KPC, OXA-48, TEM, and tetW are ubiquitous? Should tetW be present at the highest level?

We agree that contextualizing our findings within the broader research landscape is useful for demonstrating consistency and significance. Our results regarding ubiquity and relative abundance of specific ARGs align well with global trends observed in wastewater studies. As noted by the reviewer, we found that bla_{CMY}, bla_{CTX-M}, bla_{KPC}, bla_{TEM}, and tetW are highly prevalent in wastewater across the United States, with bla_{OXA-48} following closely behind. The ubiquity and high concentration of tetW is consistent with its widespread detection reported in numerous wastewater treatment plant influent globally as reported by Wang et al in their review (Distribution, sources, and potential risks of antibiotic resistance genes in wastewater treatment plant: A review; 2022). Furthermore, our finding that tetW is often present at the highest concentration among genes measured, is also consistent with the prevailing understanding of ARG distribution in wastewater environments. A review by Wang et al (Occurrence and fate of antibiotics, antibiotic resistant genes (ARGs) and antibiotic resistant bacteria (ARB) in municipal wastewater treatment plant: An overview; 2020) also identified bla_{CTX-M}, bla_{TEM}, tetW to be few of the most commonly detected ARGs in wastewater treatment plants. They also noted that tetracycline resistance genes were higher concentration than beta-lactamase genes, which is consistent with our results. While ARG concentrations can vary with geographical location, sampling time, and specific methodological approaches, the relative abundance patterns observed in our nationwide study largely correspond with established global trends in wastewater resistomes. Lastly, in addition to the previous studies already mentioned in the manuscript that link socioeconomic status to risk of AMR, a recent preprint that compared socioeconomic factors with cultured bacteria from sewersheds in Atlanta found similar correlations as our study.

To elaborate on these comparisons, we have edited the text in the Discussion section as follows:

With our extensive sampling across 163 wastewater treatment plants, we confirmed that several high-concentration targets, including bla_{CMY}, bla_{CTX-M}, bla_{KPC}, bla_{TEM}, and tetW, exhibit broad ubiquity across centralized wastewater systems in the U.S. Out of these high-concentration targets, genes like tetW that exhibit minimal variation nationwide, may not offer much insight into the resistance burden since its distribution is relatively consistent. This high prevalence and general abundance of tetracycline resistance genes is consistent with patterns previously reported in wastewater studies.^{34,35}

Also in Discussion: For example, in a recent preprint, Goetgeluck et al. cultured bacteria from 12 diverse sewersheds in Atlanta and found higher concentrations of

fluoroquinolone-resistant Enterobacterales, and third-generation cephalosporin-resistant *E. coli*, *Klebsiella*, *Enterobacter*, or *Citrobacter* spp. in sewersheds with higher proportions of crowded households, Hispanic, non-Hispanic Asian, and individuals speaking a language other than English at home, similar to significant potential determinants found in our study.²²

6. Line 125. Not all readers may be familiar with ‘z-scores’ and thus it would be useful either here or in the Materials and Methods to mathematically define how they are calculated.

We have added the mathematical formula for z-scores under the Data analysis section in Materials and Methods. The corresponding sentence now reads:

Z-scores were calculated for each target by comparing the concentration to the overall distribution across all samples, using the mean (μ) and standard deviation (σ) in the following equation: $z = \frac{x - \mu}{\sigma}$.

7. Lines 176-186. How is urbanicity defined? The SI considers the distribution of urbanicity along a scale from 0 to 100, but Figure 2 defines urbanicity using a Yes/No scale. Is >50% urban a Yes or is something closer to 90% considered urban? The clear definition needs to be provided. A similar lack of clarity also affects the authors use of the term ‘urban’ – Figure 2 defines it using a Yes/No scale, but what that means in the context of Figure S11 is unclear. How dense does an area need to be to be considered urban?

Detailed information about how urbanicity was calculated for each sewershed is described in the SI. Briefly, we obtained urban areas from the US Census Bureau and determined the proportion of each sewershed area that intersects an urban area. For data analysis, urbanicity was treated as a binary variable using a 50% cutoff as mentioned in the Methods: “The sewershed were considered urban if there was 50% or more area considered to be urban.” We added the following text to the SI and the Methods section again to avoid confusion:

Sewersheds with an urban area proportion $\geq 50\%$ were classified as urban and sewersheds with an urban area proportion $< 50\%$ were classified as non-urban for data analysis.

Reviewer #2 (Remarks to the Author):

Thanks for asking me to review this manuscript. It was a pleasure to read.

In brief, Pickering et al., sought to understand factors that may contribute (associate with) to the distribution of AMR (in particular 11 targeted ARGs) across the US by measuring their abundance at 163 separate WWTP in the US. These sites included those within 40 separate

states. They integrated this data with publicly available data on antimicrobial use and census data to look in a holistic fashion on factors that associate with AMR.

The manuscript is particularly well written and easy to follow. The use of the same colour for different regions made the manuscript easy to read. Its broad scope and relevance make it an ideal candidate for consideration of publication in Nature Communications. Perhaps the most contentious (only in some circles) is the finding that local antimicrobial prescribing (in this regulated, controlled environment) only have a modest impact. This is really important and a bit more discussion on this (what is there is great) would be fantastic given the magnitude of this global crisis.

There are several issues that should be addressed to make the manuscript more robust and easier to follow.

We are grateful for the reviewer's positive and encouraging assessment of our manuscript. We have carefully considered all suggested improvements and have addressed each point comprehensively in our revised manuscript and response to reviewers below including adding more information on the monitored sewersheds, our methods for both sampling and analysis, and the secondary variables used for clarity. In addition, we've separated *mecA* from the rest of the beta-lactamase genes throughout the manuscript and emphasized the correlational nature of our analysis, among other edits suggested by the reviewer.

Major issues.

1. Details on the sewersheds that are being monitored and when they are being assessed are key – and are lacking here.

We have added more details on the sewersheds as described below.

a. Please include a table on the nature of each of the WWTP - and the size of the population served. As the authors rightfully highlight the comprehensive nature of their study (i.e. 40 states included!) - it would be nice to see what fraction of the US population is accounted for in these samples which would allow the reader to determine how much stock to put into their projections across each county.

We have added a table (Table S3) that lists each individual treatment plant, the type of sample captured, whether the sewer systems are separate or combined, and the population served. Additionally, we have added a line to the manuscript to indicate how much of the population is being accounted for in our samples:

The total population served by the treatment plants was 24,455,284, representing 7% of the US population.

b. What are the influences of rainwater in the system? Some communities have separate storm and sewage wastewater systems - and although many do not. While 16S will somewhat mitigate this - it should be commented on.

Of the 162 plants, 21 had combined sewer systems. As a result, and because of the normalization we performed, we expect the influence of rainwater to be minimal. We have added text acknowledging this in the limitations:

Similarly, weather effects were not considered although it is possible that severe rainwater events could affect the concentrations of ARGs either due to infiltration or a dilution effect to a small proportion (n=21) of our treatment plants that had combined sewer systems (systems where stormwater and sewer water are combined).

i. The authors use the term grab samples – but also describe this as potentially being an aggregate sample collected over 1-8. Please distinguish if this was the same or different – and include in the table relative to those that provided 24-hours composite samples. Are differences observed between them based on diurnal patterns of water usage?

We have added information about sample type to Table S3 in the supporting information. As samples were not collected at different timepoints, no analysis of diurnal patterns of water usage was possible. We note that “grab samples” in this context represent samples from a primary clarifier, which have significant retention times and effectively act as composite samplers. In our previous work, we have shown that centrifuging influent to obtain wastewater solids is a suitable substitute for when primary settled solids are not available and that the two matrices correlate closely. To clarify our use of each sample type, the following edit was made in Methods:

In *Sample collection*: Wastewater treatment plant (WWTP) staff provided either “grab” samples from the primary clarifier or 24-hour composite influent samples from the headworks during a one-week period in May 2024 (Table S4 shows which WWTP collected each type and the date range of the collected samples).

In *Pre-analytical processing and nucleic-acid extraction*: Specifically, our prior research showed comparable detection sensitivity and strong correlation in ARG concentration between these sample types, with wastewater solids exhibiting slightly higher ARG concentrations on a per-mass basis.⁴⁴ This indicates that wastewater solids serve as a representative and effective matrix for assessing community-level ARG prevalence and that centrifuged influent pellets were a suitable substitute when wastewater solids were not available.

In *Pre-analytical processing and nucleic-acid extraction*: Wastewater solids were obtained from the sample by use of Imhof cones for influent samples and centrifugation for primary settled solids and then suspended in a buffer (DNA/RNA shield, Zymo Research, R1100-250, California, USA) containing bovine coronavirus...

c. Samples appear to have been connected at a single point (or week in time) - are these the same through the entire study? If not - please indicate when for each site. What are there seasonal influences?

Our study design is cross-sectional, and therefore all wastewater solids samples were collected from each of the wastewater treatment plants during a single, defined sampling

campaign conducted over a one-week period in May 2024. This concurrent sampling approach was critical to minimize temporal variability across sites and enable a snapshot assessment of community-level antibiotic resistance across the U.S. We have edited Table S4 to include the date range of sample collection for each plant and added the precise sampling time frame to the Methods section for clarity. The added text reads:

Wastewater treatment plant (WWTP) staff provided either “grab” samples from the primary clarifier or 24-hour composite influent samples from the headworks during a one-week period in May 2024 (Table S4 shows which WWTP collected each type and the date range of the collected samples).

In the results section: An average of 2.7 samples were collected per wastewater treatment plant (WWTP) over a week in May of 2024 and concentrations were averaged to obtain one representative concentration for each target per treatment plant (Table S1).

Regarding seasonal influences, we acknowledge that our cross-sectional design does not allow for a direct assessment of temporal dynamics. We have previously addressed the potential for seasonal effects in the limitations section of the Discussion. As noted, while some studies suggested limited overall seasonal trends in ARG profiles, we recognize that our snapshot approach may not capture potential delayed responses or subtle seasonal variations in specific factors. We believe that our study provides a robust national baseline given the concurrent sampling, and future longitudinal studies would be valuable to explore temporal trends in more detail. No changes were made to the manuscript.

i. While authors have argued some studies have not shown seasonal influence of AMR distribution – many, many have particularly as it pertains to respiratory viral seasons influence.

We agree that the literature on seasonality of ARGs in wastewater is complex, with some studies indeed reporting distinct seasonal patterns while others showing modest effect. Due to our study being a cross-sectional study, we cannot conclusively comment on seasonal variability of the ARGs based on the data from this study as mentioned above. We have revised the relevant section in the Discussion section discussing limitations to reflect this complexity more accurately. The edited text now reads:

Second, it is important to note that this is a cross-sectional analysis and while this design provides a valuable national baseline of ARG concentrations, it does not permit direct assessment of temporal or seasonal influences. The literature on ARG seasonality in wastewater is complex; while some studies report no obvious trends in time and only a modest effect of seasonality, others have identified significant seasonal patterns.^{36,38–41} Additionally, if any factors lead to a delayed response in the wastewater signal, as recent studies suggest could occur with antibiotic use,^{42,39} this study would not be able to assess such effects.

ii. Limited serial monitoring with only 2.7 samples per WWTP all collected (I think) within one week. The authors do not describe or include information as to whether this was done at the same time and what time this might be. While some m

As mentioned above, text was added to the Methods section for clarity and to point readers to where this information can be found in the supporting information.

d. Social variables in these WWTP require more details.

Please see our response to Reviewer 1 general comment 2. We have included additional details in the main text; complete data processing steps were previously described in the SI. As mentioned in the main text, the sociodemographic variables we selected come from the CDC's Social Vulnerability Index as social vulnerability describes the degree to which demographic and socioeconomic factors affect a community's resilience (such as the ability to respond to public health emergencies).

i. Airports - what was defined as an airport? Are these only large hubs - by some specific definition or does it include even small non-commercial strips? While definitions report # of hospitals the author's analysis is presented as presence/absence.

Regarding airports, please see our response to Reviewer 1 general comment 2. As detailed by the data source reference (Federal Aviation Administration's National Airspace System Resource Aeronautical Data Product) and in the SI, we only included large hubs (i.e., those in the "1,000,000 or more" layer from the data source: <https://hub.arcgis.com/maps/esri::usa-airports/about>). Table 1 also defines this variable as "number of major airports" to indicate that we only considered large hubs.

Regarding hospitals, we chose to analyze hospitals as a binary variable upon viewing the distribution of hospitals in our study population as stated in the Methods (Figure S14): "Number of airports and hospitals were split into groups based on the presence of the facilities, and the number of nursing homes were split by the national median number of nursing homes (median: 4)". Additionally, the national median number of hospitals per sewershed was 1. Table 1 defines how we initially characterized sewersheds whereas the text in the Methods explains how we ultimately determined to include variables for data analysis. We added the following to the SI to avoid confusion:

For data analysis, airports and hospitals were split into groups based on the presence of the facilities, and the number of nursing homes were split by the national median number of nursing homes (median: 4).

ii. Hospitals/LTC - how are hospitals defined? Are they only tertiary or quaternary sites or specifically large LTC facilities? I find it hard to imagine that any of these large catchments for WWTP don't include a small community hospital or nursing home? While these are presented as < or > 7 - what does this mean? 1 large nursing home of 350 individuals is likely more relevant than 8 small nursing homes of 40 individuals....

Please see our response to Reviewer 1 general comment 2. Locations of hospitals and nursing homes were obtained from the US Department of Homeland Security Homeland Infrastructure Foundation-Level Data (<https://hifld-geoplatform.hub.arcgis.com/>). As stated in the Methods, sewersheds were stratified by the number of nursing homes facilities present in the sewershed area. We have updated the analysis to reflect the national median number of nursing homes: less than 4 or 4 or more, irrespective of facility size. Unfortunately, not all facilities reported valid data for “population” or “beds”, so we could not take facility size into account. This limitation was added to the discussion (see Reviewer 1 general comment 2).

iii. More detail on all of these variables would help the reader.

We agree, please see our response to Reviewer 1 general comment 2. More details were added to the main text and complete information is included in the SI. All data sources are cited so the reader may explore the datasets we used at their leisure.

2. MecA is not - as stated - a beta-lactamase and should not be grouped with others. If the authors wish to continue to include it – they will need to modify several figures. Its inclusion, however, in this list as an AMR threat is a bit misleading. MecA is relevant in so far as its presence in S. aureus is a threat, however, the vast majority of its measured abundance will derive from CONS which would not be considered an AMR threat.

We thank the reviewer for pointing out this detail about *mecA*. We have removed *mecA* from the list of beta-lactamase genes and listed it out as a separate category in all our analysis and figures. Despite its distinct mechanism and prevalence in commensal coagulase-negative staphylococci (CONS), we have decided to retain *mecA* in our target list for two primary reasons:

1. Potential clinical threat through mobility - *mecA* is primarily carried on a cassette, a mobile genetic element capable of horizontal transfer. Even if its high concentration in wastewater is partially reflective of the presence of non-pathogenic CONS, it represents a substantial reservoir of this critical resistance gene. The presence of such a mobile element in the environment indicates a potential for its acquisition by susceptible *Staphylococcus aureus* strains, contributing to the emergence and dissemination of MRSA.
2. Clinical relevance and stakeholder interest - selection of *mecA* as a target gene was also informed by discussions with clinicians, who expressed an interest in understanding the wastewater levels of this specific gene due to its clinical implications.

Therefore, we believe that its surveillance in wastewater offers valuable insight into broader resistome pertinent to clinical concerns.

3. Quality control is important – but how relevant is supplemented BCoV - an RNA target well established for SARS-CoV-2 - as a control in an AMR experiment? No results are presented on this – and I suggest to remove it entirely as it does not seem to be relevant at all. If you think so

– please describe why it is important and provide data demonstrating which samples were excluded owing to the results generated.

We agree that BCoV is not an AMR-specific target; however, it was included as an internal extraction control to assess the overall nucleic acid extraction efficiency and the presence of PCR inhibition across our diverse set of wastewater samples. We followed an established protocol, cited in the manuscript, which incorporates BCoV as an extraction and RT-qPCR control. These quality control steps ensure the reliability and comparability of ARG quantification data, aligning with principles of minimal information for quantitative PCR experiments (as outlined in the MIQE guidelines). While we did not include specific BCoV quantification results in the main manuscript to maintain focus on the ARG findings, overall recovery was reported in the Methods. We believe this quality control measure significantly enhances the confidence in our reported ARG concentration. To address the reviewer's concern of relevance, we added a brief statement in the Methods section explaining the purpose of BCoV as follows:

BCoV served as an extraction and inhibition control to assess nucleic acid recovery efficiency and to identify samples with significant PCR inhibition... Based on the recovery, we concluded that there was no gross inhibition of the samples and no samples were removed due to low BCoV recovery.

4. I am unfamiliar about EPIC Cosmos and its ability to provide comprehensive antimicrobial usage data. Is this human only - or does it include animals and agriculture? Does it include hospitals or just communities? Is it represent ALL antimicrobial prescribing or just a small sampling? - please provide some background even if it is in the supplement for the reader to understand. Currently supporting citation is the company.

Epic Cosmos only records human prescriptions of antibiotics, as it represents de-identified patients records from electronic health records across the United States. In this study, we used the results of a total of 1,590,238,036 patient encounters in the time period under question to assess how often antibiotics were prescribed. We have added text to the manuscript indicating this. Please refer to Reviewer 1's General Comment 5.

a. From this data the authors have assessed beta-lactam usage. However, lumping all beta-lactam antibiotics together for assessing selection influence for AMR is likely inappropriate. Beta-lactams can be very narrow- or very broad- spectrum. What the authors have done is like lumping all things with wheels together Some with wheels might be unicycles others semi-trailer trucks and others yet are 747s.... It is not the wheel (i.e. the beta-lactam ring) that differentiates them. The authors should at the very least look at narrow (e.g. penicillin, amoxicillin, ampicillin) vs broad (amoxi-clav, amp-clav, pip-tzp, ert, mem, imi) etc separately.

We agree that lumping all beta-lactam antibiotics together may oversimplify the diverse selective pressures as the reviewer mentioned. To address this, we have revised our analysis and manuscript to categorize beta-lactam usage into two more distinct groups: 1) carbapenems and 2) penicillins and cephalosporins. This refined categorization allows us to differentiate the highly broad-spectrum Carbapenems from the more varied

spectrum of Penicillins and Cephalosporins, providing a more refined and nuanced understanding of their respective contributions to ARG prevalence. We have updated analysis and figures accordingly throughout the manuscript so that our correlation analysis and predictive modeling takes beta-lactam use and carbapenem use as two separate variables as shown in Figure 2 and 4, in addition to their corresponding figures in the SI.

5. The authors have identified the highest burden of AMR is in the NE and the S – and this is clear in Figure 1. However, figure S7 looking at overall AMR and S6 for specific targets suggest that within the NE there is considerable heterogeneity with much of this small area having a low burden of resistance. Please comment on those factors contributing -and the heterogeneity that is apparent in the data.

Figure 1 presents the mean sum of ARG z-scores directly from our empirical wastewater concentration data collected at 163 sewersheds across the U.S. This figure reflects the mean sum of all of the individual ARGs z-scores for that region. Figure S7 (now Figure S8) is a predictive map (at the county level) of overall AMR burden generated by our random forest model, which was trained on our sewershed data and relevant secondary data. The differences observed by the reviewer, particularly the heterogeneity within a region could be due to the fact that the random forest model's performance for predicting overall ARG burden (represented by z-scores) was notably poor (as noted in the discussion), evidenced by a very low R^2 value of 0.02 in our previous model and now 0.15 in our modified model. This indicates that the model struggled to accurately capture the complex, multi-faceted nature of aggregated resistance when predicting across the entire U.S. The aggregation of diverse ARGs into a single z-score likely obscured gene-specific signals, leading to reduced predictive accuracy for the overall burden. This aligns with our finding that more robust insights were gained when examining and modeling individual genes separately.

We have further clarified these points in the discussion section to emphasize the limitations of aggregated modeling. We believe that while the overall burden map provides a high-level overview, the more granular, gene-specific analyses and the direct observations from our sampling sites offer more robust insights into community-level resistance dynamics. The discussion section text now reads:

The use of z-scores to aggregate resistance burden allowed for a standardized comparison, but may obscure gene-specific insights that were obtained when examining individual genes, consistent with other summary metrics used in previous studies.^{35,36} This highlights the value of measuring individual genes and modeling their abundance separately, and underscores the difficulty of developing an overall resistance burden indicator. The random forest model also performed less effectively with z-scores representing overall resistance (**Figure S8**). Consequently, while useful for a high-level overview, the aggregated burden map should be interpreted with caution, as it may not fully capture the nuanced heterogeneity observed in the underlying gene-specific data or the localized observations from our sampling sites. This aligned with the correlation

analysis where more insights were gained by examining individual genes, underscoring a need for granular analysis to understand community-level resistance dynamics.

6. *The authors substitution of not-detected as half of the limit of detection should be cited – although this is well supported in other chemical wastewater based studies. I do fear however, that this may have influenced some of their analyses.*

As the reviewer mentioned, this is a common practice in environmental chemistry and also in wastewater-based epidemiology studies. The following citation has been added to show it has been used in previous wastewater surveillance studies:

- Kantor, R. S. *et al.* Operationalizing a routine wastewater monitoring laboratory for SARS-CoV-2. *PLOS Water* **1**, e0000007 (2022).

While there is some discussion on comparing different methods to handle non-detects in qPCR data, there is no such assessment with digital PCR yet to the best of our knowledge. Substituting half of LOD is slightly different for dPCR as it takes factors such as number of partitions assessed into account when assessing the LOD of each sample, as opposed to qPCR where the same LOD is used across all samples. To rigorously assess its potential influence on our analyses, we performed a sensitivity analysis by re-running all models and statistical comparisons using a value of zero instead of half-LOD for non-detects. We found no significant differences in our primary conclusions or model outcomes. A summary figure illustrating the results of this sensitivity analysis has been added to the Supporting Information and referred to in the main text in the Methods with the following text:

Summary of the analysis when 0 was substituted for non-detect values is shown in Figure S10.

a. As mcr was identified the least (i.e. negative in 38.3% samples) - how much significance are we seeing in the population modelling as a result of their assumption of not detected being equivalent of half the limit of detection? What if these were changed to zero? Given its strongest associations with SES factors this would be wise. This is also true for NDM (21.5% non-detects).

As part of our sensitivity analysis on the handling of non-detects, we specifically re-evaluated the associations for mcr-1 and NDM when non-detect values were replaced with zero instead of half the limit of detection. Interestingly, for both mcr-1 and NDM, this change resulted in more pronounced associations with the identified determinants if any. This is likely due to the increased range in concentration values when non-detects are set to zero, thereby amplifying the observed relationships. As a more conservative approach to avoid the assumption of absolute zero presence when a gene may be simply below the analytical detection threshold, we chose to retain the half-LOD substitution in the main analysis.

7. The authors do not (to the degree required) – distinguish between association and causation in their discussion. This would be very important particularly as it pertains to the SES factors assessed here – and some caveats through the manuscript would be wise.

We fully agree that distinguishing between association and causation is important, particularly when discussing complex socio economic factors. Please see our response to Reviewer 1 General Comment 1 on our edits to clarify this point.

a. A multivariate analysis - particularly with such a large dataset to try to disentangle the many observed associations to determine which are independent would be nice.

Please refer to Reviewer 1 Comment 1 to see our additional analysis conducted.

Minor issues.

1. The authors do a good job in the figures – but less so in the text in emphasizing that their data has been normalized for 16S rRNA.

To highlight the 16S rRNA normalization, we have revised the Methods section to clearly state that all ARG concentrations were normalized to 16S rRNA gene copies.

The added text reads: All ARG concentrations were normalized by 16S rRNA concentration to account for variations in bacterial biomass in each sample.

In addition, we have edited text throughout the results section to indicate that ARG concentrations were normalized by 16S rRNA when referring to our quantitative ARG data to emphasize the normalization step we have taken so that the text reads: Overall, most 16S rRNA normalized ARGs in wastewater were positively correlated with each other... and: While colistin resistance gene concentration normalized by 16S rRNA did not differ based on the presence of hospitals or nursing homes...

a. They do not also describe the rationale for this step

The 16S rRNA normalization was done to account for variations in bacterial biomass in each sample and is the gold standard in reporting ARG concentrations. We have added in the following citations of studies that investigate normalization of ARG concentrations to the statement above.

- Keenum, I. *et al.* A framework for standardized qPCR-targets and protocols for quantifying antibiotic resistance in surface water, recycled water and wastewater. *Crit. Rev. Environ. Sci. Technol.* **52**, 4395–4419 (2022).
- Yin, X. *et al.* Toward a Universal Unit for Quantification of Antibiotic Resistance Genes in Environmental Samples. *Environ. Sci. Technol.* **57**, 9713–9721 (2023).

b. Please ensure consistency in naming. 16S, 16s rRNA

We have gone through the manuscript to consistently use the term 16S rRNA.

2. *The authors bounce around between ddPCR and dPCR. Were both used – and if so where and when? Or was only one methodology employed.*

We clarify that our study exclusively employed droplet digital PCR (ddPCR) as our quantification method. While ‘dPCR’ was used as a general term for digital PCR technologies, we have revised the manuscript to consistently use the term ‘ddPCR’ throughout the text for clarity when referring to our experimental methods and results. The term ‘dPCR’ was only used in the introduction when presenting advantages of the broader category of digital PCR technologies.

3. *There are inconsistencies using proper formatting for genes vs proteins throughout the figures and texts.*

We confirm that our manuscript exclusively refers to antibiotic resistance genes and not protein names. To ensure consistent and proper formatting for gene names, we have conducted a thorough review of the entire manuscript and the supporting information.

4. *I think a spelling mistake here “At each split” - or each site? Line 205.*

We appreciate the reviewer’s careful attention to detail. The “split” referred to here was not a spelling mistake, but rather a technical term referring to the process within the random forest model algorithm. In random forest, “at each split” (or node) of a decision tree, a random subset of variables is considered for determining the best way to divide the data. For clarity we have edited the text at the first mention of a split to:

The random forest models presented here had one tuning parameter: the number of variables randomly sampled at each split (node), which ranged from 2 to 28.

5. *Why are the data not always presented for each of the ARG-targets in each of the figure and analyses? (ie Fig S4 only 7 of the 11 gene targets (many beta-lactamases but not all) - and these differ from Figure S4 where a different 7/11 are presented . Figure S3 has 8/11. Figure S6 11/11.*

We understand the concern about the varying subsets of genes shown in different supplemental figures. Our intention was to avoid redundancy by not presenting the same full set of 11 individual gene plots in every figure if they were included in the main manuscript, particularly given the extensive number of associations explored. However, we recognize that this approach has inadvertently led to confusion and the perception of incomplete or inconsistent data presentation. To address this, we have revised our supplemental figures to ensure that all ARG targets are consistently presented in any figure that shows individual gene-level data. The updated figures should provide a complete visual overview for each analysis, enhancing clarity and the reader’s ability to interpret the data comprehensively for every gene.

a. *If there is a rationale for the presenting only these selective examples please provide that.*

Our rationale for including a subset of the quantified ARGs in the main text was to highlight those with the most compelling or representative associations that are central to our key findings and discussion points. These figures were carefully chosen to visually support the primary narratives developed in the main body of the paper, particularly in the Discussion section, providing focused examples that exemplify the complex relations observed. The complete set of results for all ARG targets is now presented in the Supplementary Information. For every place that only presents a subset of the analysis in our main manuscript, we have added in the figure caption that the complete set is available in the SI.

Reviewer #3 (Remarks to the Author):

The authors measured 11 antibiotic resistance genes by digital droplet PCR in wastewater solids obtained from 163 wastewater treatment plants across the United States. The topic is very interesting, and the work has significant potential for impact. Further clarification on the following aspects is needed for readers to better understand its approach and potential impact.

We thank the reviewer for their time and insightful comments. We have addressed each aspect mentioned below for clarity, including adding more information on sampling and the secondary variables used in analysis and adjusting our model to include regions with similar characteristics to our training set.

Data Quality:

• Line 391: Wastewater concentrations vary significantly both diurnal and weekly. While the diurnal variation can be addressed using composite samples, the impact of weekly fluctuations remains unclear. Limited sampling (2.7 samples per plant) could introduce uncertainties. More details on the sampling and data processing are needed to justify the sufficiency of the average of 2.7 samples per wastewater treatment plant.

We acknowledge that wastewater characteristics, including ARG levels, may exhibit diurnal and weekly variations. Our study employed a cross-sectional design where samples from each of the 163 wastewater treatment plants were collected concurrently over a single week in May 2024. Within this one-week period, the participating plants provided samples on as many days as operationally feasible, resulting in an average of 2.7 samples per plant (range 1 to 4 samples). As stated in the Methods section, each time point was treated as a replicate for each wastewater treatment plant and therefore averaged to obtain a final average concentration for each target. This approach was specifically chosen to mitigate the impact of short-term (day-to-day) fluctuations and to obtain a more representative average ARG concentration for that particular week at each facility. While we agree that more intensive, longitudinal sampling over multiple weeks or seasons would provide a more complete picture of temporal variability, such an extensive sampling regimen was beyond the scope and operational feasibility of this nationwide, large-scale cross-sectional study. The objective was to provide a robust national baseline snapshot of ARG prevalence across a vast number of facilities, and obtaining multiple samples within a week was a pragmatic strategy to improve the

representativeness of each site's data within this context. The average of 2.7 samples per plant, collected over a short defined period, provides a more stable estimate compared to a single grab sample, thereby enhancing the reliability of our cross-sectional comparison across the U.S. We clarified this rationale in the Methods section and added text that reads:

Each time point was treated as a replicate for each wastewater treatment plant and therefore averaged to obtain a final representative concentration of each target. This approach was specifically designed to mitigate the influence of short-term fluctuations in wastewater characteristics, thereby providing a pragmatic method for a more robust and representative average concentration for each site during the cross-sectional sampling period.

- *Line 554: How different are the samples from multiple treatment plants within the same county? This is an important issue to examine before randomly keeping only one single value for these counties to build the RF models.*

While samples do indeed vary within treatment plants in a single county, counties represent the smallest subdivision at which secondary (demographic, hospital, and animal) data is available. We agree that the solution of keeping a single value is not ideal, as ideally one would have full coverage of the entire sewered population within the county. Lacking this, we believed that random selection would introduce less bias than averaging values across sewersheds within the counties.

- *It is not clear why only wastewater solids were analyzed and ARG in dissolved solids or the liquid fraction were excluded in this study. Can we assume the distribution of ARG in the liquid phase and suspended solids to be consistent and stable? Is ARG concentration in the wastewater solids representative of the total concentration in wastewater?*

We acknowledge that ARGs are present in both the liquid and solid phases of wastewater. Our decision to analyze wastewater solids was based on our prior work where we directly compared ARG concentrations along with other pathogen concentrations across different wastewater matrices (raw influent, centrifuged influent, and wastewater solids). Key findings from that study demonstrated that all three sample types exhibited comparable sensitivity for the detection of a broad range of ARGs but wastewater solids often yielded higher ARG concentrations on a per-mass-basis, which is advantageous for detection, especially for low abundance targets. Additionally, there was a high correlation in ARG concentrations between these different sample types across a wide spectrum of pathogen and ARG concentrations found in wastewater. This indicates that while absolute concentrations may differ, the relative patterns and presence of ARGs are largely consistent across phases. Based on these findings, we concluded that wastewater solids are a representative and effective matrix for assessing community-level ARG prevalence. Furthermore, from an operational standpoint for a large-scale, nationwide study involving numerous facilities, working with solids offers advantages in terms of sample collection, preservation, and transport logistics. We have incorporated this detailed justification, including the reference to our preprint into the

Methods section of the manuscript to provide a clear rationale for our sample matrix selection. The text reads:

Our study focused on the analysis of wastewater solids for ARG quantification. This decision was informed by previous comparative studies, including our own work, which have demonstrated that ARGs are reliably detectable and highly correlated across different wastewater matrices, including raw influent, centrifuged influent pellets, and wastewater solids. Specifically, our prior research showed comparable detection sensitivity and strong correlation in ARG concentration between these sample types, with wastewater solids exhibiting slightly higher ARG concentrations on a per-mass basis.⁴⁴ This indicates that wastewater solids serve as a representative and effective matrix for assessing community-level ARG prevalence and that centrifuged influent pellets were a suitable substitute when wastewater solids were not available.

Effect of airport presence & international travel:

- *How to interpret the effect of airport presence? A sewershed without an airport can either be a rural area or a large/mega city with the main airport just outside of the sewershed boundary. Many major cities have airports situated in neighboring municipalities or on the urban outskirts, rather than within the central urban area. This may explain why so many samples do not feature an airport (Figure 2).*

Please see our response to Reviewer 1 General comment 2 as we have added more details on our airport variable.

- *Line 271: This conclusion is not supported by the RF results, where the number of airports had a variable importance of zero, for mcr-1 and all other ARGs.*

We acknowledge that the presence of airports consistently showed a variable importance of zero across all our Random Forest models, including for *mcr-1*. We believe this observation from the Random Forest models stems from the inherent characteristics of our ‘presence of airports’ variable and the nature of the Random Forest algorithm. The ‘presence of airports’ is a relatively sparse variable in our dataset, with only a limited number of sewersheds possessing large airports considered in the study. Random Forest models, while robust for high-dimensional data, may assign low importance to variables with limited variability or very few positive instances, especially when other, more continuous or variable-rich parameters can better explain the observed variance across the majority of the dataset. In such cases, the model might consistently select alternative variables for splitting decisions, leading to a zero importance score for the sparse variable.

However, as correctly noted in our analysis, the significance of ‘presence of airports’ was identified through Wilcoxon post-hoc test with Benjamini-Hochberg correct when comparing ARG concentrations between sewersheds with and without airports. This statistical test is specifically designed to assess differences between two distinct groups and is less sensitive to overall feature space density compared to a predictive model like

Random Forest. To avoid confusion, we have decided to exclude the airport variable from our RF model. For further explanation and edits to the manuscript, please refer to Reviewer 1 General comment 3.

Model accuracy:

• *Line 214: with R^2 values ranging from 0.30 to 0.40, can the explanation that a "significant amount of the underlying variation" is captured by non-clinical indicators be fully justified? Given the relatively low model performance, more details are needed to assess whether the RF models were trained properly to achieve the best possible accuracy. Key considerations include:*

a. The quality of the ground truths: Were the wastewater samples representative of broader trends? More information on sampling strategy would help clarify this.

We thank the reviewer for outlining these key considerations. It is important to clarify that, for community-wide ARG prevalence, a direct "ground truth" in the form of comprehensive clinical surveillance data at the population level is generally unavailable. Overcoming this limitation and fragmentation of clinical surveillance, which often captures only a fraction of the circulating resistance burden in a community, is our main motivation for developing wastewater surveillance methods. Therefore, our wastewater data, rather than being compared to an external "ground truth," serve as a novel and complementary source of community-level ARG prevalence. However, to help with assessing the representativeness of our wastewater samples, we have added additional details about the sampling strategy and included it in the supporting information with a Table S4.

b. Input features selection and elimination: How were the 25 input feature selected? Was there a formal feature selection process? How to justify the appropriateness of the selected features?

To minimize any potential biases, all features were included in the modeling, with the exception of presence of airports since only a small portion of counties both in our training set and national data had airports. We selected demographic and socioeconomic factors based on the CDC's Social Vulnerability Index (SVI) because social vulnerability describes a community's ability to respond to natural or human-caused stressors. SVI variables that exhibited little variation among sewersheds (e.g., two or more races) were excluded from the analysis and subsequent model. All other features were selected based on previous studies and hypotheses regarding drivers of antibiotic resistance. Please refer to our response to Reviewer 1 General Comment 1 to see our additional analysis on input features and their co-variance.

c. Line 557: Why were antibiotic usage variables excluded from the RF models? Given their likely impact on ARG concentrations, how reliable are the variable importance results from a model that does not include these key predictors?

We have now included the antibiotic usage variables in the modeling and updated the manuscript to reflect this change. Originally, antibiotic usage variables were excluded

from the modeling as we did not have them available for the entire nation (only a subset of counties). We have since been able to extract these variables for all counties.

d. Line 204: why were the random forest models tuned using only one parameter, while there are many other parameters that can be tuned? Also, given the complex input-output relationships, how was the decision to use “only 2 variables at each split” justified?

We decided to only use the number of variables at each split as the tuning parameter as that typically has the largest impact on model performance. Due to the complex nature of the interactions, we did not see a need to reduce the depth of our trees.

The decision to use only 2 variables at each split was justified based on minimizing root mean square error. As mentioned in Methods and briefly in the Results section, we first varied the number of features at each split between 2 and 28 and found that using 2 variables at each split resulted in the smallest root mean square error. Other methods could be used, but in an effort to not bias our modeling, we chose RMSE. As this information is already present in the manuscript, no change was made.

Importance analysis:

• A major focus of this study is to understand the factors influencing ARG concentrations. This was explored through preliminary statistical analysis (Figure 2) and variable importance analysis using RF. Given the huge uncertainties with RF, why were alternative statistical/multivariate analysis techniques not considered?

Several alternate modeling methods were considered including gradient boosted random forest models, neural networks, principal component analysis and factor analysis. We chose random forest models mainly to preserve the interpretability of our models as compared to XGBoost or neural networks. We have described our rationale for choosing random forest modeling in the Methods section of the manuscript:

Based on our secondary data's characteristics, we chose random forest modeling to predict wastewater ARG concentrations in every county in the United States. Random forest modeling allows us to use data with varied distributions and nonlinear relationships, is resistant to outliers, provides interpretable feature importance, and is robust to covariance between independent variables.^{61,62}

Reviewer comments

Reviewer #2 (Remarks to the Author):

I was pleased to re-review the manuscript by Pickering et al., I am pleased to see that it is again being considered for publication – it is an impactful manuscript that is broadly representative of the US population. The authors have done an excellent job of trying to address reviewer’s comments (longer than the paper itself!) in a thorough and thoughtful manner. I am satisfied with these efforts – and the manuscript is both stronger, and easier to read. Most importantly the appropriate caveats and qualifiers are now included. A few very, very, very minor comments; Occasional areas in which nomenclature for genes is not adhered to (ie vanA in line 2019). Lines 2610-263 comparing FQ resistant Enterobacterales vs 3rd gen cephalosporin resistant (E. Coli, Kleb, Enterobacter, Citrobacter ... these account for >90% of Enterobacterales clinical isolates... this could be rephrased to appear more consistent. I recommend this for FULL ACCEPTANCE – minor areas can be changed during the page setting step.

We thank the reviewer for the positive endorsement and another round of careful review. We’ve gone through the manuscript to look for nomenclature mistakes and have rephrased the sentence mentioned for clarity as:

For example, in a recent preprint, Goetgeluck et al. cultured bacteria from 12 diverse sewersheds in Atlanta and found that sewersheds with higher proportions of Hispanic, non-Hispanic Asian, non-English speaking residents, and crowded households exhibited higher concentrations of fluoroquinolone-resistant Enterobacterales, and third-generation cephalosporin-resistant bacteria (*E. coli*, *Klebsiella*, *Enterobacter*, or *Citrobacter* spp.), similar to significant potential determinants found in our study.

Reviewer #3 (Remarks to the Author):

I appreciate the authors’ efforts to address the concerns raised during the initial round of review. However, several key issues have not been properly addressed.

Based on these comments, we have added a justification for our sampling and modeling methods in the manuscript. In addition, we have conducted a sensitivity analysis as requested by the reviewer to examine the effect of randomly selecting a sewershed to represent a county when there were multiple sewersheds in a county, in which we found that there was minimal difference in performance of the models, as described below.

For example:

- Sampling limitations: I am unconvinced by the explanation regarding the limited sampling. While I acknowledge the logistical challenges associated with this large-scale study, the core issue lies in the representativeness of the snapshot data. Collecting samples within the same week does not sufficiently address this concern—wastewater characteristics are influenced by weather conditions and local water usage patterns, which can vary significantly across locations, even within the same week. A more robust justification for the representativeness of

the snapshot data is required before the detailed results discussion can be considered acceptable. At a minimum, the limitations of the snapshot approach should be explicitly acknowledged and embedded throughout the discussion (though I don't think this alone would make the approach acceptable.)

We fully agree that environmental factors such as weather and local water usage patterns may introduce significant short-term and seasonal variability into wastewater characteristics. To directly address flow fluctuations and dilution, all target ARG concentrations were normalized to 16S rRNA gene concentrations. This effectively accounts for site-specific dilution caused by factors like storm events and local water usage, establishing our data as ARG concentrations normalized by total bacteria in each sample, allowing comparison of target concentrations across sites.

Given that our study's primary objective was to establish a robust national baseline cross sectional snapshot of ARG prevalence, we chose to focus on samples within a single, defined, one-week period. Collecting all the data within the same week minimizes the confounding effect of macro-temporal variability (i.e. seasonality) when comparing distant geographical regions. While seasonality of ARGs may exist, concurrent sampling ensures that any background seasonal effect is uniform across the dataset, making it a valid approach for establishing a national baseline for future similar time periods. The following studies have also investigated temporal variability of ARGs in wastewater and found minimal inter-day variability with both sequencing and PCR methods, suggesting that effects of micro-temporal variability are small and therefore our week-long sampling is likely to be representative of each site during this time period:

- Chau, Kevin K., et al. "High-resolution characterization of short-term temporal variability in the taxonomic and resistome composition of wastewater influent." *Microbial Genomics* 9.5 (2023): 000983.
- Sun, Shaojing, et al. "Temporal variations of antibiotic resistance genes in influents and effluents of a WWTP in cold regions." *Journal of Cleaner Production* 328 (2021): 129632.

To acknowledge the limitations of the cross sectional approach explicitly, we have added the following text to our discussion of limitations:

As such, researchers intending to use this dataset as a baseline for future studies should note that this is a cross sectional data set that does not capture temporal (seasonal) variability of ARGs.

- Random selection for RF modeling: The response regarding the decision to randomly keep a single value for counties with multiple treatment plants in the RF models remains insufficient. This decision needs to be justified by discussing other alternatives.

The decision to randomly select a single value for counties was driven by spatial resolution mismatch between our environmental data (wastewater treatment plant specific ARG concentrations related to sewersheds) and the secondary data

(aggregated at the county level). We considered the two most practical alternatives for synthesizing the multiple wastewater treatment plant values into a single-county value: calculating the arithmetic mean of all ARG concentrations from wastewater treatment plants within that county vs randomly selecting one concentration to represent the county. Both methods have their strengths but ultimately we chose to randomly select a value to minimize structural bias. While this single selected value may not be perfectly representative, the random selection process avoids introducing systematic weighting bias and maintains the original, observed value (rather than generating a synthetic mean) for the model's training process. We considered minimization of bias into our dataset more important than data sparsity, especially when dealing with complex, non-linear models. While we concur that a full census approach, which maps ARG concentration to the entire sewershed population of the county is ideal, we are limited by the publicly available resolution of the secondary data.

To confirm the reliability of selecting single sewersheds for counties represented multiple times in the dataset, a sensitivity analysis with 50 independent random trials was performed for each target gene. The analysis showed that the model's performance was consistent and highly stable (tight clustering of results). This high stability confirms that the modest predictive power is a consistent constraint imposed by the current features, suggesting that while the existing features capture some relevant information, additional unmodeled drivers of antibiotic resistance exist.

The following changes were made to the manuscript in order to provide more detail on our method. To emphasize the limitations of the matching process, we have edited our section on the challenges of matching sewershed data to county-level data to read:

Additionally, matching sewershed boundaries to counties or census tracts may have introduced inaccuracies when linking resistance data to local characteristics, especially in training our random forest prediction models.

Furthermore, we have added text in the methods section for predictive modeling our rationale for random selection of a value:

We chose this approach instead of averaging the values to minimize structural bias, as averaging would systematically and incorrectly assign equal weight to all plants regardless of their population size, which we could not correct for since wastewater treatment plants typically do not report the number of population served broken down by county boundaries. A sensitivity analysis regarding this approach can be found in the SI.

We have added the new sensitivity analysis to the manuscript in the SI under the Additional details on modeling section as follows:

Sensitivity analysis. To validate our choice of selecting one random sewershed when a county was represented more than once in the dataset, we performed a sensitivity analysis using 50 different independent trials of random selections for each target gene. Overall, the model exhibited tightly clustered predictive performance across the majority

of the target genes (**Table S9, Figure S16**). The model's predictive power, measured by the mean R², clusters around 0.26 to 0.33 for most genes with the highest average R² score achieved for *mcr-1* (0.39) and the lowest score for *bla*_{OXA-48} (0.18). The repeated trials did not lead to large swings in performance as shown by the tight clustering for each gene. The high stability for most targets confirms that this modest performance is a consistent limitation of the feature set, rather than a matter of noisy training. Therefore the features used in the model are likely capturing relevant information, but additional features are required to significantly increase the overall predictive accuracy, suggesting presence of additional drivers of antibiotic resistance that were not captured in this study and highlighting the complexity of factors that contribute to antibiotic resistance.

Table S9. Model performance stability across different randomly selected

Target	R ² mean ± standard deviation	R ² range	Number of trials
bla _{CMY}	0.28 ± 0.057	[0.181, 0.443]	50
bla _{CTX-M}	0.27 ± 0.032	[0.224, 0.355]	50
bla _{KPC}	0.271 ± 0.042	[0.189, 0.375]	50
bla _{NDM}	0.329 ± 0.043	[0.266, 0.476]	50
bla _{OXA-48}	0.182 ± 0.052	[0.091, 0.343]	50
bla _{TEM}	0.26 ± 0.046	[0.17, 0.366]	50
bla _{VIM}	0.334 ± 0.046	[0.238, 0.454]	50
mcr-1	0.39 ± 0.061	[0.262, 0.498]	50
mecA	0.261 ± 0.04	[0.186, 0.331]	50
tetW	0.26 ± 0.039	[0.201, 0.4]	50
vanA	0.317 ± 0.026	[0.259, 0.392]	50

sewershed for counties represented by multiple sewersheds

Figure S16. Model performance stability across different randomly selected sewersheds for counties represented by multiple sewersheds. The median is shown by the line inside the box with the 25th and 75th percentile represented by the lower and upper boundary of the box. Bottom and top whiskers show 1.5 x interquartile range.

- *Feature selection: The authors have modified their input feature set (e.g., removing airport presence, adding antibiotic usage variables), but there is no discussion on how these changes affected model performance or sensitivity. This, again, raises concerns about how the selected features are adequately justified.*

We removed the airport presence due to data sparsity and low variance and based on recommendations in the first round of reviewer comments. Antibiotic usage was added because that data became available nationally during the manuscript revision, and we felt it should be included as a theoretical driver of ARG prevalence. Despite being mechanistically relevant, the antibiotic usage variables did not rank highly among the overall important predictors in the final random forest models. This observation corresponds directly with our correlational analysis, which demonstrated that certain socioeconomic and demographic factors exhibited stronger associations with ARG concentrations in wastewater than the available antibiotic usage metrics. The model's low prioritization of usage variables reinforces our manuscript's conclusion that broader societal contexts are important determinants of ARG prevalence.

To maintain clarity and succinctness, we will only present the results from the final, optimized model in the main manuscript. However, for transparency, we have included more information and discussion in the SI to describe the initial feature set and compared performance metrics to the final feature set. This additional analysis is referenced in the main manuscript.

In the methods section of the main manuscript:

More details on our modeling approach can be found in the SI.

In the SI:

Additional details on modeling

Feature set. Our prediction model involved minor, strategic adjustments due to data limitations and availability. Adjustments made included: 1) sparse features, such as airport presence, were removed to mitigate the risk of overfitting to rare observations. 2) Conversely, the national antibiotic usage data became available during the project and was included to represent an important mechanistic driver of ARG prevalence. 3) to prevent extensive extrapolation to unsewered areas, the model's predictive scope was limited to U.S. counties where secondary data values fell within the minimum and maximum ranges of the training set for all variables. The impact of these feature changes on model performance is detailed in Table S8. While R^2 values for individual ARGs remained largely consistent with minor fluctuations, the R^2 or the overall summed ARG metric increased substantially, rising from 0.02 to 0.15. However, even with this increase, the total resistance metric was not able to explain as much variance as the predictive models for individual genes, reinforcing the idea that focusing on provides more insightful detail than relying on a summary metric of antibiotic resistance when there are multiple classes of antibiotics and different mechanisms of resistance involved.

Despite being mechanistically relevant, the antibiotic usage variables did not rank highly among the overall important predictors in the final random forest models (Figure 4). This observation is consistent with our correlational analysis, which demonstrated that certain socioeconomic and demographic factors exhibited stronger associations with ARG concentrations in wastewater than the available antibiotic usage metrics. The model's low prioritization of usage variables reinforces the conclusion that broader societal contexts play a key role in ARG prevalence.

Table S8. Model performance changes with adjustment to feature set.

Prediction model target	Initial R^2	Adjusted feature set R^2
bla _{CMY}	0.41	0.37
bla _{CTX-M}	0.29	0.30
bla _{KPC}	0.30	0.30

bla _{NDM}	0.31	0.37
bla _{OXA-48}	0.25	0.14
bla _{TEM}	0.26	0.28
bla _{VIM}	0.30	0.35
mcr-1	0.40	0.46
mecA	0.22	0.21
tetW	0.29	0.26
vanA	0.36	0.34
Total resistance	0.02	0.15

- *The factors influencing ARG concentrations: The decision to rely entirely on preliminary statistical analysis and RF variable importance for attribution analysis is not well-supported. There are many interpretability methods for models such as XGBoost and neural networks, and the statement that interpretability is lacking in these models is not valid and should be reconsidered. This links back to my earlier concern regarding the reliability of attribution analysis based on RF alone.*

Our decision to rely on the Random Forest model was based on a preliminary analysis that was not included in the manuscript and a combination of technical factors better suited to the specific characteristics of our environmental dataset. We initially evaluated several alternative modeling methods, including linear regression, support vector regression, generalized additive models, gradient boosted random forest, and neural networks. The random forest model provided the best overall performance in terms of relationship between predicted and actual ARG values when we split our data into training and test data. The gradient boost models specifically did not show dramatic improvement in performance and exhibited evidence of spatial overfitting where the resultant map of predicted values largely mirrored state boundaries rather than general trends), making them unsuitable for generalization. The neural network performed similarly to RF but showed significantly decreased stability and significantly higher variance in results when training. Given the limited size of our dataset, we concluded that using a simpler model like RF minimizes the risk of overfitting compared to more complex architectures like Neural Networks and Gradient Boost models, which typically require significantly larger datasets to achieve stable and generalized results, especially when RF provided better stability and comparable performance.

Additionally, as previously noted, RF is highly robust to multicollinearity among independent variables, which may be present in socioeconomic and demographic factors, preventing unstable coefficient estimates that would affect linear models and certain generalized linear models. RF also does not make assumptions about the

underlying distribution of the data, allowing it to capture non-linear dependencies, which we expect between ARG prevalence and various factors, more effectively. Lastly, while we acknowledge that other models can be interpreted, the native RF variable importance derived directly from feature splits across the different decision trees provides a stable and easily communicated measure of attribution strength, which helps in translating our findings into clear public health insights based on county-level features.

We have revised the methods section to articulate a more comprehensive technical justification, emphasizing RF's robustness to data characteristics and its stability with limited sample size to support our methodological choice. The text now reads:

Random forest modeling allows us to use data with varied distributions and nonlinear relationships, is resistant to outliers, provides interpretable feature importance, and is robust to covariance between independent variables. Given our limited sample size for the cross-sectional analysis, the random forest model minimizes the risk of overfitting often associated with more complex architectures (such as neural networks and gradient boost models) when trained on small datasets.

Reviewer #4 (Remarks to the Author):

The authors present a well constructed response to the initial reviewer feedback. I do not have any additional concerns. This is a very well written article that balances the nuances of WBS with the need to understand drivers in human communities.

We thank the reviewer for their time in going through both the manuscript and response to the initial reviewer feedback.

Below, please find the *comments of the mediating referee in black italic type and the original comments from reviewer 3 in grey italic type*. Our responses are **indented plain blue text**. The comments have been grouped by the structure of the original comments provided by reviewer 3 to be addressed.

I appreciate efforts made both by authors and other reviewers. Here I provide my response, as a mediating referee for validity and clarity, mainly regarding the responses to reviewer #3 below.

We appreciate the reviewer's time and thoughtful consideration of both our manuscript and other reviewers' comments. Since the mediating reviewer's suggestions were addressed in our previous revision, this document clarifies where those specific edits and responses can be found in the current text.

I appreciate the authors' efforts to address the concerns raised during the initial round of review. However, several key issues have not been properly addressed. For example: - Sampling limitations: I am unconvinced by the explanation regarding the limited sampling. While I acknowledge the logistical challenges associated with this large-scale study, the core issue lies in the representativeness of the snapshot data.

→I recognize the representativeness issue, given the sampling was done in a limited time frame, thus do not fully represent all four seasons in this study. However, as the authors addressed, the purpose of this study is not to investigate seasonality, but rather identify AMR patterns and the underlying drivers without confounder (i.e., temporal variation or seasonality) – I think the authors' approach, sampling sewage within a limited time frame looks reasonable. The authors reasonably well address the potential issue.

Collecting samples within the same week does not sufficiently address this concern—wastewater characteristics are influenced by weather conditions and local water usage patterns, which can vary significantly across locations, even within the same week.

→It is partly true – the potential temporal variability may exist, but it can also be overshadowed by a bigger overarching factors (i.e., social indicators in this study). The model's predictability may improve with the incorporation of temporal variability. However, the core rationale remains unchanged, as the authors already demonstrated that social vulnerability indicators significantly correlate with ARG concentrations using the available data.

A more robust justification for the representativeness of the snapshot data is required before the detailed results discussion can be considered acceptable. At a minimum, the limitations of the snapshot approach should be explicitly acknowledged and embedded throughout the discussion (though I don't think this alone would make the approach acceptable.)

→I agree with this point. The authors have already acknowledged the limitation in the manuscript (L361-388). Additionally, the authors explain why such limitation still wouldn't harm the core conclusion in this study, then how to address it the way forward.

We thank the reviewer for their assessment. As the reviewer noted, our sampling method is justifiable for the conclusions drawn regarding the correlation between social vulnerability indicators and antibiotic resistance genes in wastewater. Given that the reviewer agrees the current manuscript effectively addresses the potential

limitations of this approach, no further changes were made to this section.

Random selection for RF modeling: The response regarding the decision to randomly keep a single value for counties with multiple treatment plants in the RF models remains insufficient. This decision needs to be justified by discussing other alternatives.

→To me, it is unclear why the reviewer #3 thinks that randomly keeping a single value for counties with multiple treatment plants in the RF models remain insufficient. The reviewer can further clarify and provide the detailed rationale why he/she/they think that way.

→From my perspective, random selection of a single WWTP for each county makes sense as it will enable to fairly compare different counties with the same degree of freedom, further test variability of ARG concentration across different plants within the same site.

We appreciate the reviewer's perspective on this point. We agree that random selection allows for a consistent comparison across counties while maintaining equal degrees of freedom. Furthermore, the sensitivity analysis included in our Supporting Information during the previous revision specifically addresses the robustness of this selection method. As the reviewer found this approach reasonable and the existing analysis accounts for potential variability, no further changes were made.

- Feature selection: The authors have modified their input feature set (e.g., removing airport presence, adding antibiotic usage variables), but there is no discussion on how these changes affected model performance or sensitivity. This, again, raises concerns about how the selected features are adequately justified.

→This comment is reasonable – the authors may provide their justification on their feature selection.

We acknowledge the reviewer's point and would like to clarify that these justifications were incorporated during the previous round of revisions. To maintain the manuscript's focus, the main text presents the final optimized model and feature set; however, we expanded the Supporting Information to include a section titled "Additional details on modeling" that includes a detailed discussion of the initial feature set and how our feature selection influenced model sensitivity and performance. As this documentation is already present in the current SI, no further changes were made.

- The factors influencing ARG concentrations: The decision to rely entirely on preliminary statistical analysis and RF variable importance for attribution analysis is not well supported. There are many interpretability methods for models such as XGBoost and neural networks, and the statement that interpretability is lacking in these models is not valid and should be reconsidered. This links back to my earlier concern regarding the reliability of attribution analysis based on RF alone.

→The authors may provide a reason why they chose RF model over other alternatives.

We appreciate the reviewer's suggestion to clarify our model selection. As detailed in our previous revision, we performed preliminary analysis evaluating multiple architectures. While we acknowledge the advancements in interpretability for complex

models, our data showed that Random Forest provided the most stable performance. Additionally, given our specific sample size, we prioritized RF to minimize the risk of overfitting, a common issue with high-capacity models like Neural Networks on smaller datasets. Because the manuscript's method section was previously updated to highlight RF's robustness and suitability for this data scale, no further changes were made.

ROUND 2 REVIEWER 5 ATTACHMENT:

I appreciate efforts made both by authors and other reviewers.

Here I provide my response, as a mediating referee for validity and clarity, mainly regarding the responses to reviewer #3 below.

I appreciate the authors' efforts to address the concerns raised during the initial round of review. However, several key issues have not been properly addressed. For example:

- Sampling limitations: I am unconvinced by the explanation regarding the limited sampling. While I acknowledge the logistical challenges associated with this large-scale study, the core issue lies in the representativeness of the snapshot data.

→ I recognize the representativeness issue, given the sampling was done in a limited time frame, thus do not fully represent all four seasons in this study. However, as the authors addressed, the purpose of this study is not to investigate seasonality, but rather identify AMR patterns and the underlying drivers without confounder (i.e., temporal variation or seasonality) – I think the authors' approach, sampling sewage within a limited time frame looks reasonable. The authors reasonably well address the potential issue.

Collecting samples within the same week does not sufficiently address this concern—wastewater characteristics are influenced by weather conditions and local water usage patterns, which can vary significantly across locations, even within the same week.

→ It is partly true – the potential temporal variability may exist, but it can also be overshadowed by a bigger overarching factors (i.e., social indicators in this study). The model's predictability may improve with the incorporation of temporal variability. However, the core rationale remains unchanged, as the authors already demonstrated that social vulnerability indicators significantly correlate with ARG concentrations using the available data.

A more robust justification for the representativeness of the snapshot data is required before the detailed results discussion can be considered acceptable. At a minimum, the limitations of the snapshot approach should be explicitly acknowledged and embedded throughout the discussion (though I don't think this alone would make the approach acceptable.)

→ I agree with this point. The authors have already acknowledged the limitation in the manuscript (L361-388). Additionally, the authors explain why such limitation still wouldn't harm the core conclusion in this study, then how to address it the way forward.

- Random selection for RF modeling: The response regarding the decision to randomly

keep a single value for counties with multiple treatment plants in the RF models remains insufficient. This decision needs to be justified by discussing other alternatives.

→ To me, it is unclear why the reviewer #3 thinks that randomly keeping a single value for counties with multiple treatment plants in the RF models remain insufficient. The reviewer can further clarify and provide the detailed rationale why he/she/they think that way.

→ From my perspective, random selection of a single WWTP for each county makes sense as it will enable to fairly compare different counties with the same degree of freedom, further test variability of ARG concentration across different plants within the same site.

- Feature selection: The authors have modified their input feature set (e.g., removing airport presence, adding antibiotic usage variables), but there is no discussion on how these changes affected model performance or sensitivity. This, again, raises concerns about how the selected features are adequately justified.

→ This comment is reasonable – the authors may provide their justification on their feature selection.

- The factors influencing ARG concentrations: The decision to rely entirely on preliminary statistical analysis and RF variable importance for attribution analysis is not well-supported. There are many interpretability methods for models such as XGBoost and neural networks, and the statement that interpretability is lacking in these models is not valid and should be reconsidered. This links back to my earlier concern regarding the reliability of attribution analysis based on RF alone.

→ The authors may provide a reason why they chose RF model over other alternatives.